# The effect of hydrophobic glassy organic material on the cloud condensation nuclei activity of particles with different morphologies

Ankit Tandon[1, 2], Nicholas E. Rothfuss[1], Markus D. Petters[1]

[1]Department of Marine Earth and Atmospheric Sciences, North Carolina State University, Raleigh, 27695-8208, USA
[2]School of Earth and Environmental Sciences, Central University of Himachal Pradesh, Dharamshala, 176215, India

*Correspondence to*: Markus D. Petters (markus_petters@ncsu.edu)

**Abstract.** Particles composed of organic and inorganic components can assume core-shell morphologies. The kinetic limitation of water uptake due to the presence of a hydrophobic viscous outer shell may increase the critical supersaturation required to activate such particles into cloud droplets. Here we test this hypothesis through laboratory experiments. Results show that the viscosity of polyethylene particles is $5 \times 10^6$ Pa s at 60 °C. Extrapolation of temperature dependent viscosity measurements suggests that the particles are glassy at room temperature. Cloud condensation nuclei (CCN) activity measurements demonstrate that pure polyethylene particles are CCN inactive at diameters less than 741 nm and 2.5% water supersaturation. Thus, polyethylene is used as proxy for hydrophobic glassy organic material. Ammonium sulfate is used as proxy for hygroscopic CCN active inorganic material. Mixed particles were generated using coagulation of oppositely charged particles; charge-neutral polyethylene-ammonium sulfate dimer particles were then isolated for online observation. Morphology of these dimer particles was varied by heating, such that liquefied polyethylene partially or completely engulfed the ammonium sulfate. Critical supersaturation was measured as a function of dry particle volume, particle morphology, and organic volume fraction. The data show that kinetic limitations do not change the critical supersaturation of 50 nm ammonium sulfate cores coated with polyethylene and polyethylene volume fractions up to 97%. Based on these results, and a synthesis of literature data, it is suggested that mass transfer limitations by glassy organic shells are unlikely to affect cloud droplet activation near laboratory temperatures.

## 1 Introduction

Atmospheric aerosol sometimes consists of mixed particles with inorganic and organic components present in comparable mass fractions (Cruz and Pandis, 2000; Pöschl et al., 2010; Prenni et al., 2003). Most of the inorganic compounds, commonly salts, are hygroscopic. The organic fraction is composed of a large number of compounds, many of which remain unspeciated. The contribution of the organic fraction to cloud condensation nuclei (CCN) activity depends on their hygroscopicity. The effective hygroscopicity parameter (Petters and Kreidenweis, 2007) of organic compounds varies between 0 and ~0.3, which is based on laboratory, field, and modeling studies (e.g. Prenni et al. 2007; Gunthe et al. 2009; Mikhailov et al., 2009; Chang et al. 2010; Massoli et al. 2010; Cappa et al. 2011; Mei et al. 2013; Pajunoja et al., 2015; Petters et al. 2016; Nakao 2017). A

small subset of species constituting the organic fraction, e.g. long chain n-alkanes with carbon nuber 16 and above and their fatty acids, have very low water affinity (Jacobson et al., 2000; Jimenez et al., 2009; Petters et al., 2016). The organic fraction can thus be divided into hydrophobic and hydrophilic compounds. Hydrophobic compounds do not take up water and do not contribute to CCN activity.

Particles containing both organic and inorganic compounds may assume different phase states and morphologies. The

phase state of dry organic matter may be crystalline, amorphous solid or semisolid, or liquid. When formed from drying of aqueous solutions droplets, the organic phase often partially or fully encapsulates the inorganic core (e.g. Ciobanu et al., 2009; Freedman et al., 2010; Reid et al., 2011; Krieger et al., 2012; Stewart et al., 2015; Altaf and Freedman, 2017). Bigg (1986) first suggested that organic compounds located at the particle surface may kinetically retard water uptake and thus reduce the particle's ability to promote CCN activity. This reduction in CCN activity may occur through several pathways. First, the mass

accommodation coefficient of water may be reduced (e.g. Chuang et al., 1997; Nenes et al., 2001). Second, slow dissolution kinetics may reduce the number of dissolved molecules in the aqueous solution (Asa-Awuku and Nenes, 2007; O'Meara et al., 2016). Finally, high viscosity may slow diffusion of water (e.g. Zobrist et al., 2011) and thus affect hygroscopic growth under subsaturated conditions (Mikhailov et al., 2009), especially at cold temperature (Berkemeier et al., 2014; Petters et al., 2019). Measurements of the diffusion coefficient of water through hydrophilic viscous matrices (Price et al., 2015), measurements of

the equilibration time scale of glassy hydrophilic organic compounds with water at high RH (Rothfuss et al., 2018), and measurements of the interplay between viscosity, hygroscopic growth, and CCN activity (Pajunoja et al., 2015) suggest that hydrophilic liquid or hydrophilic amorphous solid organic matrices are unlikely to hinder droplet activation. The main reason for this behavior is that small amounts of water uptake plasticize the organic phase, which in turn removes diffusion2 limitations. Plasticization, however, does not occur in hydrophobic organic substances, which therefore are more plausible

candidate compounds to kinetically limit water uptake and droplet activation.

Laboratory studies have tested the influence of hydrophobic organic coatings on hygroscopic growth and CCN activity (Abbatt et al., 2005; Cruz and Pandis, 1998; Garland et al., 2005; Hämeri et al., 1992; Nguyen et al., 2017; Ruehl and Wilson, 2014; Xiong et al., 1998). Sometimes, but not always, a hydrophobic organic layer kinetically limits water up-take by the inorganic core and retards hygroscopic growth or CCN activity (Abbatt et al., 2005; Nguyen et al., 2017; Ruehl and Wilson,

2014; Xiong et al., 1998). These studies are now briefly reviewed. Significant work has been performed on characterizing the transfer of small gas molecules between an aqueous phase and the vapor phase separated by an organic film (McNeill et al., 2013). The review here is limited to transfer of water vapor. Studies summarizing results for $N_2O_5$, HCl, $HNO_3$ and other small molecules are reviewed elsewhere (McNeill et al., 2013).

Studies at subsaturated RH have been performed by several investigators. Hämeri et al. (1992) show that coatings of

tetracosane (solid alkane), octanoic acid (liquid carboxylic acid), or lauric acid (solid carboxylic acid) do not prevent NaCl particles from deliquescing. Garland et al. (2005) reported that the deliquescence relative humidity for ammonium sulfate (AS) particles coated with palmitic acid was the same as for pure AS for particles with low organic mass fraction (~20%). For particles with higher mass fraction (~50%) of palmitic acid, full deliquescence required higher relative humidity. Xiong et al.

(1998) experimentally examined the effects of organic films on the kinetic growth rate of ultrafine sulfuric acid aerosols in relation to film thickness and particle diameter. They reported that the monolayer films of lauric and stearic acid retard the hygroscopic growth rate of sulfuric acid aerosols. This reduction in hygroscopic growth was 20% for a 6 monolayer (14.4 nm) thick coating of lauric acid on 40-120 nm particles and relative humidity (RH) up to 85%. The slopes of hygroscopic growth factor-RH curves decrease significantly with increasing number of monolayers and residence time. However, for films ≥14.4 nm, the growth factor-RH curve remains the same regardless of coating thickness. Ruehl and Wilson (2014) measured the droplet size formed at 99.9% RH on submicrometer particles. Ammonium sulfate particles coated with palmitic and stearic acid did not take up water as expected suggesting an accommodation coefficient of $10^{-4}$. Forestieri et al. (2018) report similar observations for NaCl particles coated with palmitic acid. Particles exceeding 80% organic fraction had reduced humidified droplet size at 99.88% RH, which was attributed to kinetic limitations. Gorkowski et al. (2017) created core-shell structures with secondary organic materials comprising the shell and found no diffusion limitations for water (and other substances) at humid conditions.

Studies examining the effect of organic coatings on CCN activation also show mixed results. Cruz and Pandis (1998) investigated the effect of organic coatings on the CCN activation of AS particles coated with glutaric acid or dioctylphthalate (DOP). The CCN activation behavior of a particle having an inorganic core with organic coating was predicted by Köhler theory and applying the Zdanovskii, Stokes, and Robinson (ZSR) assumption that the equilibrium water content of the mixed particle is additive. Particles with coatings of DOP constituting at least 70% mass of the mixed particle did not hinder activation. In contrast, Abbatt et al. (2005) observed a complete deactivation of AS particles that were thickly coated (> 13 nm) with stearic acid, although thinner coatings and other substances had no effect. Nguyen et al. (2017) reported that NaCl particles coated with palmitic and stearic acid lead to deviations from the ZSR rule consistent with kinetic limitations to water uptake. Unsaturated acids did not show any deviation from the ZSR rule. Forestieri et al. (2018) investigated the CCN activity of NaCl particles coated with oleic acid and mixtures of myristic acid/oleic acid and palmitic/oleic acid and found no deviation from the ZSR rule that would be indicative of kinetic limitations.

The most commonly used materials applied in coating studies are n-alkanes and fatty acids. These compounds are hydrophobic and CCN inactive (Petters et al., 2016). The phase state of these molecules is assumed to be either liquid or solid based on the bulk phase of the substance at room temperature. In most studies the proposed mechanism responsible for the hypothesized kinetic limitation has been a reduced mass accommodation coefficient, which hinders the kinetic rate of droplet growth. The mass accommodation coefficient is defined as the fraction of molecules that stick to the particle upon collision with the droplet. Related are kinetic limitation due to diffusional transfer through the organic material, which is controlled by the viscosity of the coating. Viscosity ranges from $10^{-3}$ Pa s for liquids to > $10^{12}$ Pa s for glassy substances (Reid et al., 2018). According to the Stokes-Einstein relationship the characteristic time required for water molecule to diffuse through a 10 nm thick coating is 69 s, $6.9 \times 10^4$ s, and $6.9 \times 10^7$ s for viscosity of $10^6$, $10^9$ and $10^{12}$ Pa s, respectively. Although it is known that the Stokes-Einstein relationship may not hold for water and high viscosity (Chenyakin et al., 2017; Price et al., 2014), increased viscosity still slows diffusion.

The above cited studies did not quantify the viscosity of the coating. This study is unique in this aspect. Polyethylene (PE) is used as a proxy for the organic material. A Dimer Coagulation, Isolation and Coalescence (DCIC) technique (Marsh et al.,
2018; Rothfuss and Petters, 2016) is used to generate mixed particles composed of PE/PE and PE/AS dimer particles. Viscosity and CCN activity of pure PE particles is quantified. Dimer morphology is varied from agglomerate to core-shell morphology. Critical supersaturation for these mixed particles is measured as a function of dry particle volume, particle morphology, and organic volume fraction. Atmospheric implications of the experiments are discussed.

## 2 Methods

The basic methodology for dimer particle preparation has been presented in detail elsewhere (Marsh et al., 2018; Rothfuss and Petters, 2016). Briefly, two monodisperse particle populations of opposite charge are generated using differential mobility analyzers (DMAs) operated with positive and negative polarity power supplies. The streams are merged and particles are given time to coagulate. Coagulated dimer particles from particles carrying +1/–1 charge are charge neutral and transmit through an electrostatic filter. The terms "monomer particle" or monomer are used to refer to particles transmitted by a single DMA and
the term "dimer particle" or dimer are used to refer coagulated particles. These particles are available for further manipulation and measurement.

Figure 1 summarizes the specific setup that was used in this study. Ammonium sulfate particles were generated using a constant output atomizer (TSI 3076) that was supplied with 0.1 % w/w AS solution using a syringe pump operated at 40 μL min$^{-1}$ flow rate. The particles were dried using two Nafion dryers (Permapure PD-50T-24MSS) and charge neutralized using
a $^{210}$Po source. Monodisperse negatively-charged 50 nm AS particles were selected using the first DMA operated with sheath-to-sample flow ratios of 5 L min$^{-1}$:1 L min$^{-1}$ and connected to a positive polarity power supply (TSI-DMA1, TSI 3071). Monodisperse AS particles prepared in this way are nearly spherical with a dynamic shape factor of 1.02–1.06, depending on diameter and drying rate (Mikhailov et al., 2009; Zelenyuk et al., 2006).

Polyethylene particles were generated by homogeneous nucleation from vapor. A few beads of a PE standard (Restek,
Polywax-850) were heated above its melting point (107 °C) in a round bottom, two-neck glass flask. Dry particle free N$_2$ boiled off from a Dewar was supplied trough the center neck. Nitrogen was used to prevent oxidation. A polydispersion of particles formed from homogenous nucleation and Brownian coagulation. Particles were size selected using a second DMA (TSI-DMA2, TSI 3071) also operated with sheath-to-sample flow ratios of 5 L min$^{-1}$:1 L min$^{-1}$ but using a negative voltage power supply to select positively charged particles. The polydisperse PE particles are likely glassy at room temperature (discussed
further below) and heavily agglomerated. Therefore, the PE particles were passed through two sintering loops. The first sintering loop is after the initial cooling of the PE aerosol, the second after TSI-DMA2. Within the sintering loop the aerosol is heated to >90 °C, which lowers the viscosity sufficiently for the particles to relax into solid, near spherical particles.

The outflows from TSI-DMA1 and TSI-DMA2 were merged into three sequentially arranged 0.3 L capacity coagulation chambers (27 s average residence time). The flow was split after the chambers with 0.6 L min$^{-1}$ being passed through an

electrostatic filter to remove all charged particles. The charge neutral particles then passed through a temperature-controlled loop with residence time 5 s. The elevated temperature liquefied the PE such that dimer particles changed from a dumbbell morphology into to a spherical morphology. Particles exiting the conditioner were charge equilibrated using a [210]Po radiation source. The size distribution of coagulated particles was measured using a radial DMA (RDMA, Zhang et al., 1995) interfaced with a condensation nuclei (CN) counter (TSI 3020) and a streamwise thermal-gradient continuous flow CCN counter (Roberts and Nenes, 2005) as particle detectors. The CN and CCNc were operated at a total flow rate of 0.3 L min$^{-1}$, respectively. The CCNc and RDMA were operated using a sheath-to-sample flow ratio of 10:1 and 2 L min$^{-1}$:0.6 L min$^{-1}$, respectively. The RDMA was configured to measure the size distribution in scanning mobility particle sizer (SMPS) mode (Wang and Flagan, 1990), scanning voltage from 3000 to 100 V (145 – 26 nm) over 210 s. The coagulation chamber, electrostatic filter, and RDMA are housed inside a temperature-controlled box that can be varied between –20 °C to +40 °C. The temperature of the box is set to a baseline temperature where dimers do not relax into spherical shape and is referred to as the system temperature. Relative humidity at the system temperature was less than 3%, much drier than conditions where ammonium sulfate dimers were observed to relax into a spherical shape due to water uptake in Rothfuss and Petters (2016).

Since the melting point of AS (280 °C, Lide and Haynes, 2009) is much higher than that of PE (107 °C) the liquid PE is expected to partially or fully engulf the AS core. The assumed morphology of a liquid droplet on a spherical solid substrate depends on the solid/liquid, solid/vapor, liquid/vapor tensions, as well as the tension of the three phase contact line (Iwamatsu, 2016). The equilibrium morphology of this system can be (1) core-shell morphology (complete wetting), (2) core-shell morphology with uneven coating thickness (cap shaped droplet), (3) partially engulfed morphology (cap shaped droplet), or (4) dumbbell shaped morphology (complete drying). Parenthetical terms correspond to those used by Iwamatsu (2016). The line tension term becomes increasingly important at smaller scales. Line tension drives the system toward core-shell morphology. There is insufficient information about the AS/PE system at 95 °C to theoretically evaluate the equilibrium morphology. However, the liquid/vapor tension of polyethylene melts is 0.034 J m$^{-2}$ (Dettre and Johnson, 1966; Owens and Wendt, 1969) and in principle a low liquid/vapor tension will favor the spreading of liquid on a high-energy surface. Reid et al. (2011) argue that core–shell morphologies are favored for a liquid phase surrounding a solid crystalline core. Freedman et al. (2010) show that ammonium sulfate particles coated with palmitic acid (produced through a coating oven), remain in core-shell structure. Palmitic acid and liquid PE both have low liquid/vapor interfacial tension. Palmitic acid and PE are both dominated by CH$_2$ functional groups and thus the solid/liquid tension and line tension terms for these two substances should be similar.

The particles are cooled down to system temperature after they relax into spherical shape in the temperature-controlled loop. Based on the above we argue that the final morphology is a core-shell structure. However, we cannot dismiss the possibility that particles assumed a partially engulfed morphology. That morphology may be either due to equilibrium shape or arise during the annealing of the particle. Thus, whether the coating of annealed PE is of uniform thickness is unknown. Scanning Electron Microscopy with Energy Dispersive X-ray spectroscopy of PE/AS dimer particles with larger diameters

was attempted. However, the analysis was inconclusive due to insufficient resolution resulting from poor conductivity of the prepared samples and necessary limits in maximum electron beam intensity to ensure that the PE did not melt.

## 2. Experiment Types

### 2.1.1 Viscosity Measurement

Viscosity of pure PE particles was measured as described in previous publications (Marsh et al., 2018; Rothfuss and Petters, 2016, 2017). For this experiment, PE/AS and PE/PE dimers are prepared by feeding AS or sintered PE particles to TSI-DMA1 and sintered PE particles to TSI-DMA2. The temperature inside the thermal conditioner is continuously scanned from 25 °C (system temperature) to 95 °C over a period of 284 min. The mode of the size distribution is monitored by the RDMA-CN/CCNc instruments. The analysis of the data follows the same steps described previously and only given in abridged form. First, the raw size distribution is fitted to a lognormal distribution function to find the peak diameter. Then, the particle geometry factor is determined (Rothfuss and Petters, 2016)

$$\xi = \frac{3}{D_{uc}/D_c - 1}\left(\frac{D_p}{D_c} + \frac{D_{uc}/D_c - 4}{3}\right) \tag{1}$$

where $\xi$ is the particle geometry factor, $D_p$ is the SMPS peak diameter, $D_{uc}$ the fully uncoalesced and $D_c$ is the fully coalesced dimer diameter. The diameters $D_{uc}$ and $D_c$ are determined from scans near the system temperature and scans near 95 °C, respectively. Next, the fitted mode diameters are binned into 3 K intervals (Marsh et al., 2018) and the mean geometry factor is derived for each bin. These data are then fitted to a logistic curve

$$\xi = 1 + \frac{3}{1 + exp[-k(T - T_r)]} \tag{2}$$

where $k$ represents the steepness, $T$ is the measured temperature of the conditioning loop, and $T_r$ is the relaxation temperature representative of the midpoint of the logistic curve. The shape factor is converted to viscosity, $\eta$ via a lookup table of modified Frenkel sintering theory (Pokluda et al., 1997; Rothfuss and Petters, 2016). Conditioner residence time, monomer diameter, and surface tension assumed in the conversion are $t = 5$ s, $D_{mono} = 50$ nm, and $\sigma = 0.034$ J m$^{-2}$, respectively. The assumed surface tension is based on measured values for polyethylene melts (Dettre and Johnson, 1966; Owens and Wendt, 1969). The inferred viscosity is only weakly dependent on the assumed surface tension value (Marsh et al., 2018; Rothfuss and Petters, 2016). Particle shape factors vary between $1 \leq \xi \leq 4$, which map to a narrow range of viscosity between $10^{-5}$ Pa s and $3 \times 10^{-6}$ Pa s for the selected conditioner residence time, monomer diameter, and surface tension. The inferred temperature dependent viscosity over this range is fitted to a modified Vogel-Fulcher-Tammann (VFT) equation (Rothfuss and Petters, 2017).

$$log_{10}\eta = A + \frac{B}{T - T_0} \tag{3}$$

where $A$, $B$, and $T_0$ are coefficients obtained from a least-square fit. By convention (Debenedetti and Stillinger, 2001), the glass transition temperature, $T_g$, corresponds to $\eta = 10^{12}$ Pa s. It is obtained by extrapolation of the VFT equation. Similarly, the temperature of the transition where $\eta = 5 \times 10^6$ Pa s, $T_c$, is computed from equation (3). This value is computed to aid comparison of transition temperatures from similar measurements made with this experimental setup.

### 2.1.2 Pure Compound CCN Measurement

The CCN instrument was calibrated using size-resolved CCN measurements. Size-resolved measurements of pure PE were also obtained. Ammonium sulfate was used to calibrate the CCN instrument supersaturation, which is set by the streamwise thermal gradient. Gradients between 6 and 20 °C were used. Ammonium sulfate experiments were performed using the system described in Petters and Petters (2016). Dried, charge-neutralized particles are passed through a DMA (TSI 3080) operated with sheath-to-sample flow ratio of 9 L min$^{-1}$:1.3 L min$^{-1}$. Particle concentration is measured using a condensation particle counter (TSI 3771) and the CCN instrument. The instrument is operated in scanning particle sizer mode with voltage downscan from 10kV to 10V. Activation diameter was obtained by a fit to the data and mapped to supersaturation (SS) using water activity from the Extended Aerosol Inorganics Model (E-AIM, Clegg et al., 1998). The resulting relationship between SS and temperature gradient is shown in the Supporting Online information.

The CCN activity of sintered PE was measured using a similar setup. A high-flow DMA column (Stolzenburg et al., 1998) operated at 9:2 L min$^{-1}$ flow ratio was used. The high-flow column was used to access larger particle diameters. The CCN instrument was operated at a gradient of 20 °C and a total flow rate of 1 L min$^{-1}$ to maximize the instrument supersaturation. First, a calibration with AS particles was attempted but particles activated at all sizes. Another calibration material, glucose was used. Glucose was selected because it is relatively less CCN active than AS (Petters and Petters, 2016; Rosenørn et al., 2006), forms near spherical test particles (Suda and Petters, 2013), and water activity vs. composition is known from bulk data (Miyajima et al., 1983).

### 2.1.3 Morphology CCN Measurements

Morphology CCN experiments proceeded on dimers with the thermal conditioner at room temperature (dumbbell morphology) and the thermal conditioner at ~95 °C (spherical morphology). Experiments were performed for AS and PE monomers as well as PE/PE and PE/AS dimers with different volume fractions. The CCNc was operated ramping nominal temperature gradients from 7 to 16 °C (0.30 to 0.66% SS) in discrete steps. At each step 3 SMPS scans were performed. For PE/AS dimers with different volume fractions (shell thickness), 50 nm diameter AS monomers were coagulated with either 60 or 80 or 100 or 120 nm diameter PE monomers. These dimers were analyzed using the same method but limiting the nominal temperature gradients between 10 and 15 °C (0.42 to 0.62% SS).

The mobility diameter of the dimer depends on the particle morphology. From previous experiments with similar setups we found that the mobility diameters of uncoalesced dimers of equal size are between 1.04 and 1.1 times larger than the mobility dimeter of coalesced dimers (Marsh et al., 2018; Rothfuss and Petters, 2016, 2017). The dry particle masses are

identical in the coalesced and uncoalesced states. Furthermore, as shown below, the PE itself is CCN inactive and does not
contribute to the solute effect. Thus, the activation of PE/AS dimers is solely controlled by the AS monomer mass, which is
unchanged in all experiments, and the kinetics of water transfer to the activating drop. The latter factor may differ with
morphology and by extension mobility diameter. Therefore, the activated fraction was determined as follows. The monomer
or dimer size and CCN response functions were fitted to a lognormal distribution function. The activated fraction (AF), i.e.
the ratio of CCN to CN, was taken at the mode diameter of the size distribution. The activated fraction was then calculated for
each scan at a particular supersaturation. A cumulative Gaussian distribution function (CDF) was used to fit the AF vs. SS
relationship. The activation supersaturation $SS_{50}$ corresponding to the SS at which AF equals to 0.5 was considered to be the
critical supersaturation (Petters et al., 2009).

## 3    Results

### 3.1  Pure compound CCN experiments

Raw data for the pure compound CCN experiments are provided in the Supporting Online Information. For PE monomers no
activation was observed at the largest diameter for which sufficient particle counts could be generated (741 nm) and thermal
gradient in the CCN instrument (20 °C). The exact supersaturation corresponding to the thermal gradient is unknown because
most of the reference glucose particles of the smallest selected particle size activated at this setting (23.4 nm). Taking glucose
$\kappa$=0.17 based on CCN measurements (Petters and Petters, 2016), this implies that SS > 2.5%.  The combination of SS = 2.5%
and $D$ = 741 nm corresponds to a state above the Kelvin condition where insoluble but wettable particles activate ($\kappa$ = 0, Petters
and Kreidenweis, 2007). Polyethylene is hydrophobic and has a contact angle with water between 79 and 111° (Boulange-
Petermann et al., 2003; Dann, 1970; Gotoh et al., 2000; Owens and Wendt, 1969). The absence of measurable CCN activity
under the selected conditions is consistent with nucleation theory that includes contact angle (Fletcher, 1962; Mahata and
Alofs, 1975). The results show that in mixed particles that are composed of AS and PE, the PE will not contribute dissolved
solute. Therefore, a 50 nm dry AS particle and 50 nm dry AS particle mixed with PE are expected to activate at the same
critical supersaturation provided that the contribution of PE to the wet particle volume at the point of activation remains small.

### 3.2  Viscosity Experiments

Two experiments were performed; temperature induced relaxation of PE/PE dimers, and relaxation of PE/AS dimers.
Results from the two experiments are graphed in Figure 2. The graphs are included to illustrate the data reduction procedure
through Eqs. (2) and (3). Fitted parameters from these experiments are summarized in Table 1. The main conclusions from
these experiments are as follows. At $T > 80$ °C, the shape of the particles no longer changes, and the particle is fully coalesced.
At those temperatures $\eta < 10^5$ Pa s. Extrapolation of the VFT curve shows that at $T \sim 20$ °C the estimated viscosity for PE is
$10^{12}$ Pa s, which corresponds to a highly viscous, potentially glassy state. This extrapolation method has been shown to predict
$T_g$ within ±10 °C for sucrose and citric acid (Marsh et al., 2018; Rothfuss and Petters, 2017). Estimated $T_c$ and $T_r$ values

between the two experiments are within 4 °C, consistent with prior results that show that relaxation of a viscous liquid around a solid particle produces similar results than the merging of two liquefying spheres (Rothfuss and Petters, 2017). The precise morphology of the fully coalesced PE/AS dimer is unknown. The change in mobility diameter for the PE/PE and PE/AS experiments was 5.96 and 5.88 nm, respectively. For PE/PE particles, this is consistent with expected dynamic shape factors for rods and the final diameter of the coalesced particles is spherical. The similarity of the shift suggests that PE/AS particles are also spherical, with liquid PE coating the AS at high temperature.

### 3.3 Morphology CCN Experiments

Figure 3 shows a typical morphology CCN experiment. The curves correspond to the SMPS scan using the CN and CCN as detector. The left panels correspond to measurements with AS monomers. For this experiment, TSI-DMA2 and the electrostatic filter were turned off. Thus, the experiment corresponds to a tandem DMA experiment with a second neutralizer placed in line. The resulting size distribution is trimodal over the range shown. Theoretical analysis and experimental verification of the size distribution produced by this tandem DMA configuration is provided elsewhere (Petters, 2018; Wright et al., 2016)[1]. Briefly, the center peak is dominated by singly charged particles transmitted by TSI-DMA1, the peak to the right is dominated by doubly charged particles that have undergone charge re-equilibration and the peak to the left is dominated by singly-charged particles transmitted by TSI-DMA1 that have acquired two charges during the charge re-equilibration step. At dT = 16K (SS = 0.66%) the CCN and CN distributions agree, indicating that particles of all sizes activated. Conversely, at dT = 8K (SS = 0.34%) none of the nominal 50 nm AS monomers activated, while most of the larger particles did. At dT = 12K (SS = 0.50%) an appreciable fraction of the 50 nm AS monomers activated.

The middle panels in Figure 3 correspond to measurements with 50 nm PE/50 nm AS dimers in their uncoalesced state (dumbbell morphology). The measured size dimer distribution is also trimodal, although for reasons different than those of the tandem DMA experiment. The shape of the dimer size distribution has been analyzed elsewhere (Petters, 2018; Rothfuss and Petters, 2016). Briefly, the center peak is dominated by dimers formed from +1/–1charged particles transmitted by TSI-DMA1 and TSI-DMA2. The peak to the right is dominated by dimers formed from coagulated +2/–2 charged particles. The peak to the left is dominated by monomers that lost their charge on transit between either size-selection DMA and the electrostatic filter. Note that at the mode of the center peak, the solute volume of AS is identical to that of the monomer experiment shown in the left panel. Consequently, the center peak is used for analysis. The ratio of CN and CCN at the mode diameter of the center peak for dT = 8, 12, and 16K are very similar to that of the monomer experiment. The only difference between the middle panel and right panel in Figure 3 is the temperature of thermal conditioner. The center mode corresponding to the +1/–1 dimers is reduced by 4.4 nm (Table S8), indicating the change in particle morphoplogy while holding the number of AS and PE molecules comprising the particle constant. The dimers in their coalesced state represent the core/shell morphology.

---

[1] Theoretical analysis of the relevant response functions reported in this work is provided in Notebook S8. Hygroscopicity Tandem DMA and Notebook S10. Dimer Coagulation and Isolation, which are supplements to Petters (2018).

Assuming a uniform coverage, the thickness of the PE shell is 6.5 nm. Again, the ratio of CN and CCN at the mode diameter of center peak for dT = 8, 12, and 16K are very similar to that of the monomer experiment.

Figure 4 shows the relationship between AF and SS at the mode diameter for 50 nm AS monomers, uncoalesced 50 nm AS/50 nm PE dimers and fully coalesced 50 nm AS/50 nm PE dimers shown in Figure 3. Here the discrete supersaturation values correspond to steps in dT = 1K, i.e. dT = 8, 9, … , 16K (0.34 to 0.66% SS). The fitted activation spectra for the three
experiments are virtually identical. Although there is some scatter in the AF data, the activation occurred at the same dT in all three cases. This implies that the CCN activation is controlled by only the AS core. Addition of hydrophobic PE to the particle has no effect, irrespective of particle morphology.

The experiment was repeated with 50 nm AS and 60, 80, 100 and 120 nm size PE dimers to vary the nominal coating thickness. These experiments are summarized in Figure 5. The volume fraction of PE varied from 50% to 96.7%, corresponding
to nominal coating thickness between 6.5 and 36.4 nm. The dimer data were obtained in the coalesced state and the monomer data are shown for reference. Again, no significant differences in the activation supersaturation are observed. In some experiments, the activated fraction did not approach unity at high supersaturation. For experiments with disparate diameter of AS and PE, e.g. 50 nm AS and 120 nm PE, the dimer diameter will approach that of the larger monomer. Since a small fraction of decharged monomers also transmits (Rothfuss and Petters, 2016, Petters, 2018), the distribution contains some fraction of
pure PE particles that will not activate. Thus, the lack of 100% activation at high SS is likely an artifact and not due to particle morphology. However, it is also possible that those particles are composed of PE/AS and have assumed a core-shell morphology that is sufficient to hinder activation.

## 4   Discussion

Question about how organic coatings influence CCN activity date back to Bigg (1986), who attributed poor aerosol-to-CCN
closure to organic coatings. The two principle mechanisms of the delay are poor mass accommodation and diffusional limitation through the organic matrix (e.g. Chuang et al., 1997, Nenes et al., 2001, Zobrist et al., 2011). Either effect also implies some influence of particle morphology on CCN activation. Core-shell structures might be shielded by the organic while agglomerates or homogeneous mixtures are not.

Hygroscopic organic compounds that dissolve will not shield inorganic substances, even if they are initially glassy.
Dissolution kinetics in crystalline substances occur on time scales faster than exposure time in CCN instruments (a few seconds) or cloud updrafts (few minutes) (Asa-Awuku and Nenes, 2007). This is corroborated by CCN experiments with sparingly soluble hygroscopic compounds that activate at their deliquescence RH, e.g. succinic or adipic acid. In their most pure state, these compounds activate according to theoretical prediction from solubility (Bilde and Svenningsson, 2004; Christensen and Petters, 2012; Hings et al., 2008; Hori et al., 2003; Kreidenweis et al., 2006), which implies no kinetic
limitation to deliquescence and dissolution for crystalline solids at the time scale of CCN experiments. Amorphous glassy particles correspond to a less ordered state than crystalline solids. One might therefore expect a lower barrier to dissolution.

Rothfuss et al. (2018) measured the condensation kinetics of water on amorphous glassy hygroscopic organic supermicron particles and found only marginal delays due to high starting viscosity. Extrapolation to submicron scales yields equilibration times ~100 ms, too fast to affect CCN experiments. The cited fundamental arguments are also consistent with the finding that water soluble and/or hygroscopic organic compounds have not shown growth delays or activation delays in coating studies (Cruz and Pandis, 2000; Garland et al., 2005; Hämeri et al., 1992). Based on this evidence, hygroscopic glassy organic substances can be ruled out to affect CCN activity through growth delays, irrespective of particle morphology.

Hydrophobic liquid organic substances may shield the inorganic core through mass accommodation effects. The studies discussed in the introduction (Abbatt et al., 2005; Nguyen et al., 2017; Xiong et al., 1998) suggest that thick coatings of lauric, stearic or palmitic acid are needed to achieve noticeable growth delays or deviation from ZSR mixing. The results by Nguyen et al. (2017) show a remarkable difference between saturated and unsaturated fatty acids, with saturated fatty acids hindering water transfer. It is known that the barrier efficiency increases with increasing organic chain length and packing density (McNeil et al., 2013). Packing density increases with attractive forces between molecules; the surface active fatty acids can bind on the hydrophobic tails and hydrophilic end groups. Lauric (C12) and palmitic (C16) acid are also macroscopic solids at room temperature due to their melting points of 44 and 63 °C, respectively (Lide and Haynes, 2009). These values are 63 and 44 °C cooler than that of the PE standard used here (107 °C). Assuming a similar shape of the VFT curve for fatty acids and PE, this implies that the viscosity of amorphous palmitic acid coatings at room temperature is $< 10^6$ Pa s. Some retardation based on viscosity is therefore expected for the fatty acids, although the viscosity is orders of magnitude lower than that of the glassy PE. The experiments here were motivated to generate core-shell structures that are both glassy and hydrophobic to investigate if the combination of the two effects will modify CCN activity. Before discussing possible reasons why no decline in CCN activity was observed, the nature of hydrophobic glassy particles is discussed.

Organic compounds become viscous and glassy due to the presence of functional groups. Rothfuss and Petters (2017) rank the sensitivity of viscosity to functional group addition from most to least sensitive as carboxylic acid (COOH) ≈ hydroxyl (OH) > nitrate ($ONO_2$) > carbonyl (CO) ≈ ester (COO) > methylene ($CH_2$).  With the exception of nitrate groups, functional groups that are responsible for high viscosity also promote hygroscopicity (Petters et al., 2016; Suda et al., 2014). To our knowledge, multifunctional nitrated organic compounds are not prevalent or rare in the atmosphere. Therefore, the only source of hydrophobic-glassy-particles are high molecular weight weakly functionalized hydrocarbons. The PE source used here had a nominal molecular weight of 850 Da, corresponding to ~60 $CH_2$ groups. The carbon number may have changed during heating via thermal decomposition. Whether or not this occurred cannot be determined from the available data. Regardless, the obtained PE particles had an estimated viscosity of $10^{12}$ Pa s at room temperature. The exact carbon number of the PE particles here is not important, addition or subtraction of a few $CH_2$ molecules does not significantly alter viscosity or hygroscopity (Petters et al., 2016; Rothfuss and Petters, 2017). Note, however, that atmospherically relevant hydrophobic organics may have double bonds, include aromatic rings (e.g. polycyclic aromatic hydrocarbons), and may include some functional groups other than $CH_x$.  Nonetheless, the PE particles used here are used as a proxy for hydrophobic and glassy compounds. What then are plausible reasons that the PE model did not affect CCN activity?

The experiments in Figure 5 can be used to bound the water diffusivity. A nominal coating thickness of 36.4 nm corresponds to a particle composed of 96.7% organic material by volume. Scaling analysis implies that the apparent diffusivity of water through the shell is $D > \sim 10^{-16}$ m$^2$ s$^{-1}$, much larger than values calculated from the Stokes-Einstein relation. This estimate assumes a uniform coating. In practice, the uniformity of the coating is unknown. At 95 °C, the estimated viscosity of PE is $< 10^5$ Pa s (Figure 2). The resulting particles are spherical. If we imagine the system as effectively liquid, the surface tension forces will lead to a completely engulfed core with uniform coating. After exiting the conditioning loop, the coating anneals. It is plausible that small fissures form in the curved shell, which would reduce the packing density and allow water to penetrate. It is also plausible that the resulting coating is not of uniform thickness due to annealing forces resulting in the core to be pushed off center. In the extreme case a partially engulfed morphology may result, which has been documented to occur in particles with slow drying rates (Altaf and Freedman, 2017; Nandy and Dutcher, 2018). Another possibility is that the known breakdown of the Stokes-Einstein equation (Chenyakin et al., 2017; Price et al., 2014) is strong enough to result in diffusion rates through glassy PE that exceed the apparent $D > \sim 10^{-16}$ m$^2$ s$^{-1}$. Indeed the measured diffusivity of water through low density PE films is $\sim 10^{-14}$ m$^2$ s$^{-1}$ (Wang et al., 2011). Thus, any of the mentioned effects, cracks, non-uniform coating thickness, or faster than expected diffusion, could have contributed to, or fully explain the lack of shielding observed in this study. Note that the saturated fatty acids for which delays have been observed have viscosity $< 10^6$ Pa s, which is low enough to allow viscous flow at the scale of submicron particles, and consequently the formation of a tightly packed viscous shell without cracks. This might suggest that the optimal barrier for shielding particles from water would be for substances that have a narrow range of viscosity where the coating is viscous enough to slow water transport but not too viscous to prevent flow around the particle.

These findings have important atmospheric implications. The permeability of polymers to small molecules is well known. For example, Nafion membrane humidifiers are widely used in the community for drying or humidifying aerosol flows (and are permeable to water vapor). Permeability depends on the polymer composition, its molecular weight, temperature, and the number and type of crosslinks (George and Thomas, 2001). A key question is what types of oligomeric or polymeric substances could form atmospheric hydrophobic glassy coatings, and by what process. Three are imagined here. Long-chain fatty acid coatings may form on sea-spray particles during the bubble-bursting process (Tervahattu, 2002). Some of these will initially be liquid and might turn glassy upon cooling or functionalization via heterogeneous aging. Weakly oxidized oils in the C$_{18}$-C$_{40}$ range are emitted by diesel engines (Sakurai et al., 2003). These compounds may form coatings on pre-existing atmospheric particles perhaps through an evaporation-oxidation-condensation mechanism (Robinson et al., 2007) and also become glassy upon cooling or functionalization via heterogeneous aging. A third possibility is the irreversible oligomerization of weakly functionalized low molecular weight organic compounds forming hydrophobic compounds that then deposit on the surface of inorganic particles during drying (Altaf and Freedman, 2017; Nandy and Dutcher, 2018), followed by turning glassy during cooling. It is unclear how common and important these processes might be in the atmosphere. To our knowledge single particle studies on dried aerosol do not show core-shell structures as the dominant particle type (Laskin et al., 2012; Li et al., 2010; Piens et al., 2016). Regardless, the imagined processes producing hydrophobic glassy coatings undergo drying and cooling

cycles that will be susceptible to the same issues reported here: cracks, non-uniform coating thickness formed during drying or annealing, partially engulfed equilibrium morphology, and faster than expected diffusion through hydrocarbon films. We therefore conclude that mass transfer limitation by glassy organic shells is unlikely to affect cloud droplet activation in the majority of cases at temperatures prevalent in the lower atmosphere. Extension of this result to temperatures in the upper free troposphere where low temperature slows diffusion may require further experimentation.

Altaf et al. (2018) report experiments that suggest that dry particle morphology impacts the activation diameter. These experiments were performed with particle mixtures composed of 50/50 w/w mixtures of AS/succinic and AS/pimelic acid particles. Particle morphology was varied by changing the drying rate; a slow drying rate produces particles with partially engulfed morphologies while a fast drying-rate produces particles with homogeneous morphology. The authors report that the partially engulfed morphologies activate at a smaller diameter than the homogeneous morphology. This result is opposite from

the one reported here, where no effect of particle morphology on CCN activity was observed. The experiments here and by Altaf et al. (2018) differ in two key aspects. First, organic material between the two studies is different. Polyethylene is hydrophobic and does not contribute to the solute effect. Succinc and pimelic acid do. Succinc and pimelic acid also slightly lower surface tension of the air/aqueous solution interface, while PE does not. Second, the amount of solute is controlled differently in this study. Here, dimer morphology is produced by coagulation with fixed AS monomer size. In contrast, Altaf

et al. (2018) used atomized mixtures, followed by drying and size selection. Altaf et al. (2018) do not have a quantitative theory to predict the observed effect of particle morphology but they propose potential explanations. One proposed explanation is that pimelic acid serves heterogeneous catalyst, the other proposed explanation is a liquid-liquid phase separation mechanism coupled with surface tension lowering similar to the mechanism proposed by (Ovadnevaite et al., 2017). Neither of these effects is applicable to the PE model system. However, we point out that the methodology used here can be used to create

core-shell structures with many combinations of inorganic cores and organic coatings to further investigate effect of particle morphology on CCN activation for core-shell structures with organic shells that may lower surface tension.

## 5    Conclusions

Dimers composed of ammonium sulfate (AS) and polyethylene (PE) and PE and PE were prepared. The dimers were used to determine the temperature dependence of the viscosity of the generated PE particles by measuring the conditions where dimers

coalesce and relax into spheres. The viscosity of PE was $5\times10^6$ Pa s at 60 °C. Extrapolation of the temperature dependence suggests that the PE particles have viscosity $10^{12}$ Pa s near room temperature. Seven hundred nanometer diameter PE particles were CCN inactive at 2.5% supersaturation consistent with a contact angle of PE with water exceeding 79°. The AS/PE dimers were used to measure the critical supersaturation of the particles in dumbbell morphology formed during coagulation and core-shell morphology formed after shape relaxation. No difference in activation supersaturation was observed up to nominal shell

thickness of 36.4 nm. An increase in supersaturation is expected if water diffusion was governed by the Stokes-Einstein relation. The apparent water diffusion through the hydrophobic plastic is orders of magnitude faster than predicted from the

Stokes-Einstein relation. Potential explanations are cracks formed during annealing, non-uniform coating thickness, formation of partially engulfed morphologies, or fast diffusion of small molecules through polymer membranes. It is argued that processes that may form glassy hydrophobic organic shells on atmospheric particles will result in similar imperfect shielding of hygroscopic cores. However, particles with thick coatings of some, but not all fatty acids are an exception to the preceding claim (Abbatt et al., 2005, Nguyen et al., 2017, Forestieri et al., 2018). Water transfer will be less hindered in hydrophilic glassy organic materials due to the plasticizing effect of water on dissolving organic compounds. Based on literature data, particles comprising hydrophobic glassy organic materials fully coating inorganic cores are not ubiquitous in the atmosphere. Furthermore, timescales of humidification are shorter in CCN instruments than in atmospheric updrafts. Therefore, our experiments suggest that, near laboratory temperatures, mass transfer limitation by glassy organic shells is unlikely to affect cloud droplet activation.

## 6    Data availability

All data underlying the figures are available from the Online Supplement.

## 7    Supplement link (will be included by Copernicus)

## 8    Author contributions

MDP designed the experiments. AT and NE carried out the experiments. AT and MDP prepared the manuscript with contributions from all co-authors.

## 9    Competing interests

The authors declare that they have no conflict of interest.

## 10    Acknowledgements

This research was funded via Department of Energy, Office of Science grant DE-SC 0012043. AT acknowledges Central University of Himachal Pradesh for granting study leave to carry out this work at North Carolina State University.

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

**Table 1: Tabulated values for viscosity experiments. Symbols not defined above: *ΔT* is the 95% confidence interval of *T*ᵣ determined from the fit, and *RH* is the relative humidity inside the coagulation chamber.**

| System | $T_c$ [°C] | $T_g$ [°C] | $T_r$ [°C] | $k$ [-] | $\Delta T$ [°C] | RH [%] | $A$ [-] | $B$ [-] | $T_0$ [K] |
|---|---|---|---|---|---|---|---|---|---|
| | | | Equation 2 | | | | Equation 3 | | |
| PE/AS | 64.5 | 23.9 | 67.7 | 0.19 | 3.8 | 2.9 | 0.7 | 519 | 251 |
| PE/PE | 60.2 | 20.7 | 63.4 | 0.19 | 2.8 | 2.3 | 1.08 | 456 | 252 |

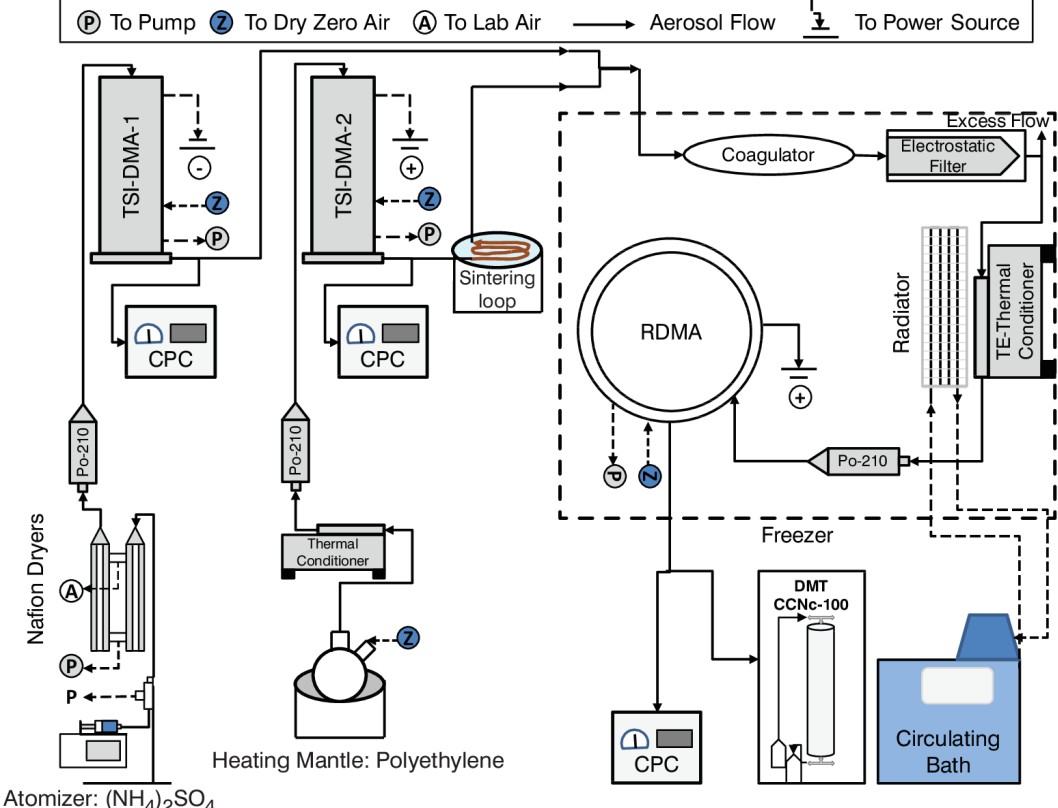

**Figure 1: Schematic of the Experimental setup (DMA: Differential Mobility Analyzer, RDMA: Radial Differential Mobility Analyzer, CPC: Condensation Particle Counter, DMT CCNc-100: Cloud Condensation Nuclei Counter, Po-210: Polonium 210 radiation source).**

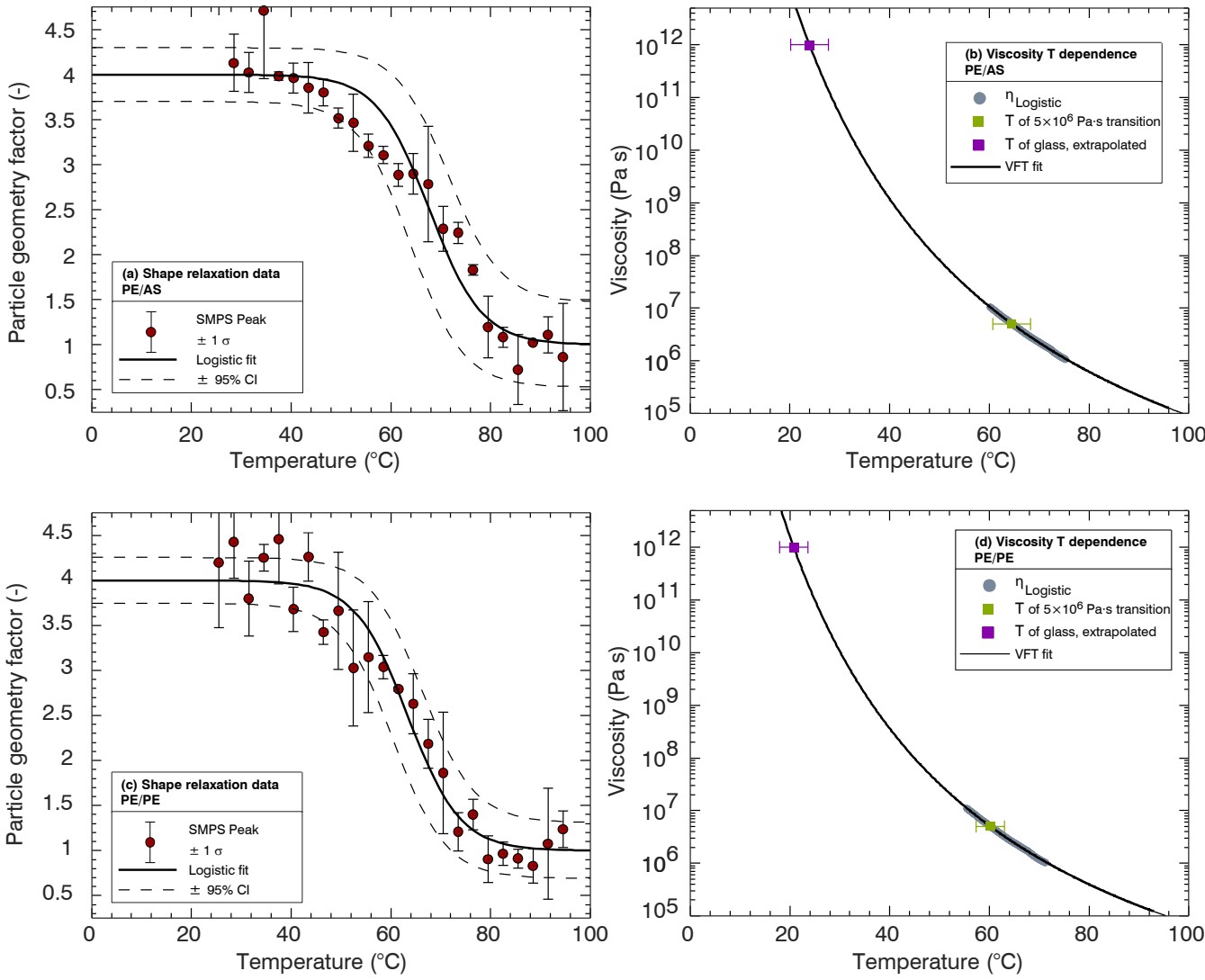

**Figure 2.** Left plots: shape relaxation data for PE/AS (a) and PE/PE (c) 50nm/50nm dimers. Symbols are the mean geometry factor derived from the temperature binned mode diameter peaks. Vertical bars are the standard deviation. Dashed lines correspond to the 95% confidence interval of the relaxation temperature derived from the fit. Right plots: temperature dependence of viscosity for PE/AS (b) and PE/PE (d) experiments. The solid line gives the VFT fit. Grey circles correspond to values where $1 > \xi > 4$. The green and purple symbols correspond $\eta = 5 \times 10^6$ and $10^{12}$ Pa s, respectively. Horizontal error bars correspond to the 95% confidence interval of the relaxation temperature derived from the fit. Parameters corresponding to the logistic and VFT fits are given in Table 1.

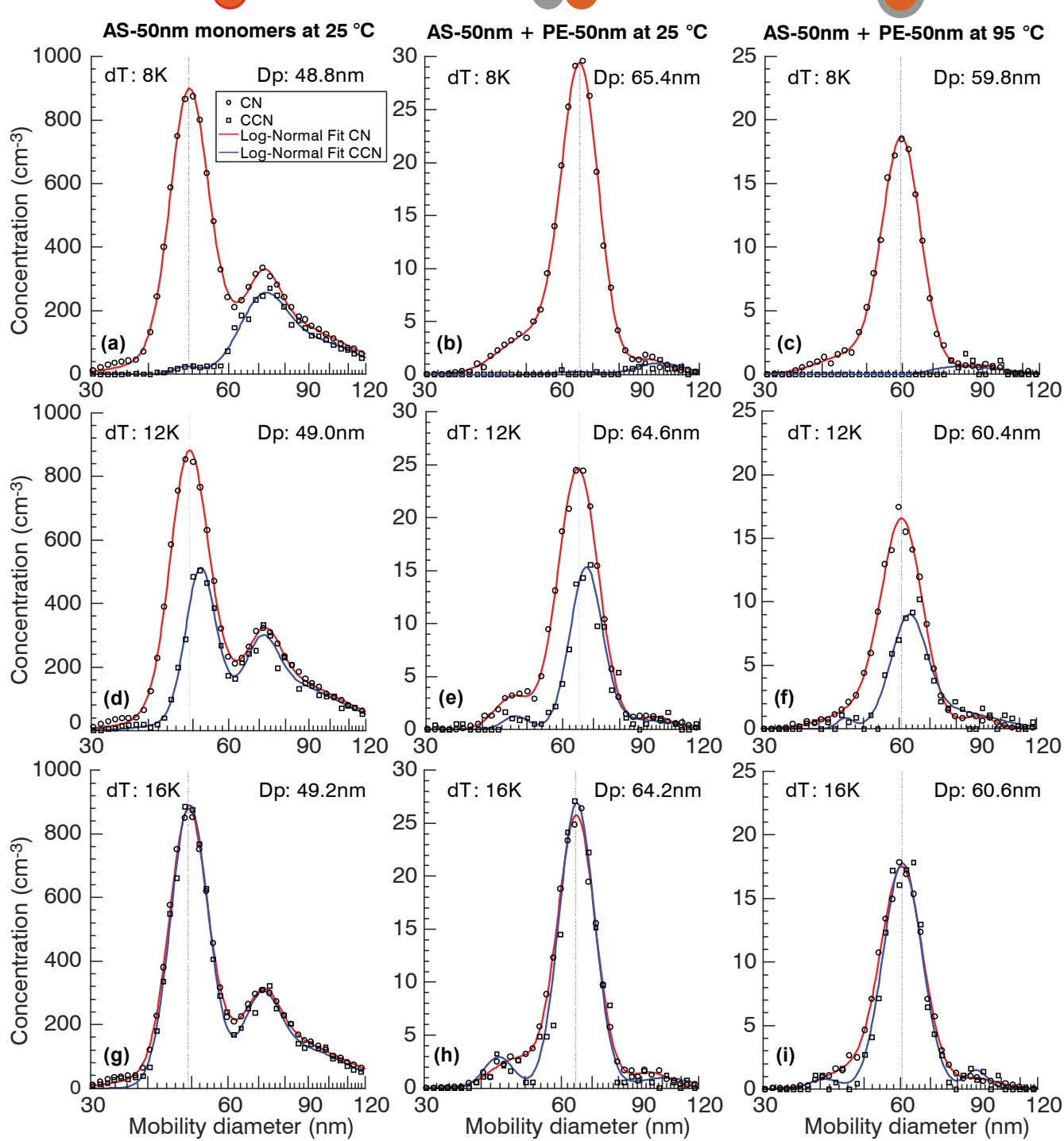

Figure 3: CN and CCN size-distributions at three different CCNc column temperature gradients corresponding to water supersaturations measured after the thermal conditioning of particles.

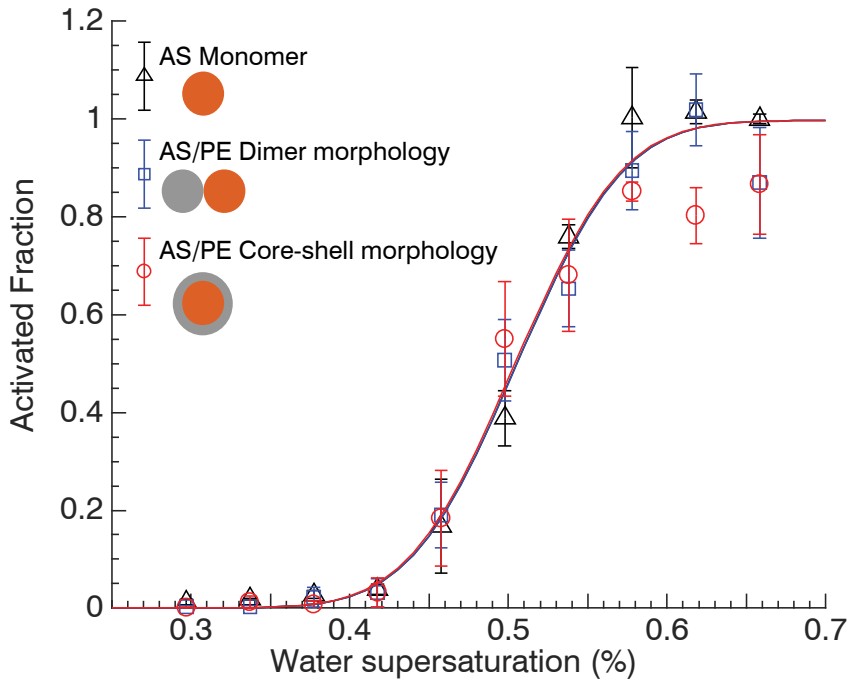

**Figure 4: Activated fraction determined at mode diameter of ammonium sulfate monomers and mixed ammonium sulfate/polyethylene particles with dumbbell and core-shell morphology at different water supersaturation. Lines correspond to a fit to the data.**

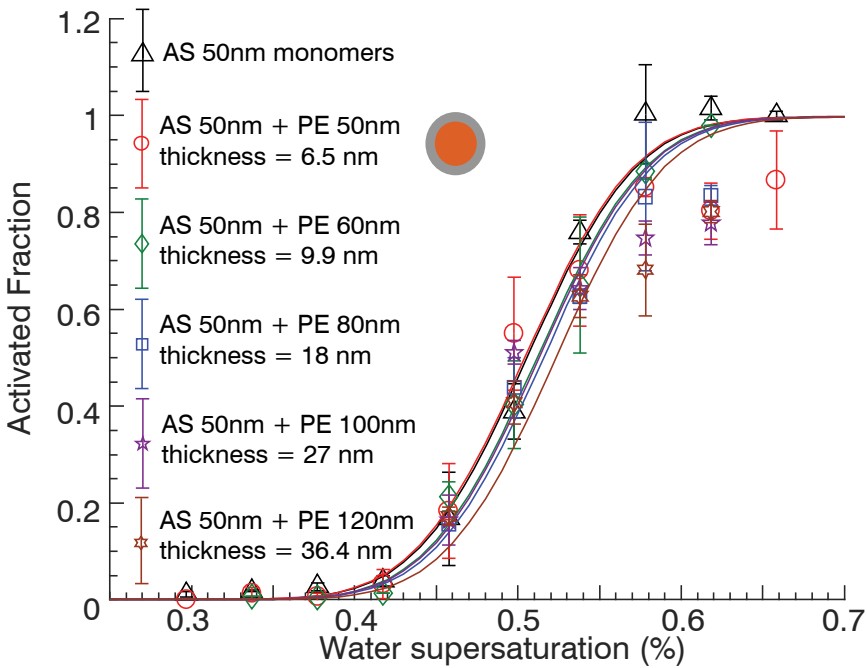

**Figure 5: Activation properties of ammonium sulfate monomers and mixed ammonium sulfate/polyethylene particles with core-shell morphology with varying coating thickness. Error bars correspond to ±1 standard deviation associated with activated faction estimated from three separate scans.**