# Peer review of "The effect of hydrophobic glassy organic material on the cloud condensation nuclei activity of particles with different morphologies"

_Atmospheric Chemistry and Physics, 2018_

## Referee Comment (RC1) · Anonymous Referee #2 · 30 Sep 2018

In the study at hand, Tandon and co-workers investigate the role of hydrophobic organic material on the cloud condensation ability of ammonium sulfate (AS) particles by coating them with polyethylene (PE). In this careful study with innovative design, no effect of the coatings was observed, rendering hydrophobic organic coatings as uninfluential under the investigated conditions. The paper is well-written, the topic is of high relevance to the Atmospheric Chemistry and Physics community and fits well within the scope of Atmospheric Chemistry and Physics.

Regarding the title, I have two issues with the use of "internally mixed" here.

[Figure]

1. This paper really rather looking at the effect of organic coatings on inorganic particles (admittedly, by creating an internally mixed particle) instead of the effect of organic coating on internally mixed particles.

2. While it is a set phrase in the aerosol community, it might paint the picture of hydrophobic substances being "mixed in" with inorganic substances.

There are not big issues, but the title threw me off slightly in the beginning and I wonder if this addition is needed.

The biggest critique for the relevance of this work must be, as the author's state themselves in their discussion, the relevance of PE coatings for atmospheric particles. PE is indeed very different to the most common organic material in the atmosphere, Secondary Organic Material (SOM). The authors argue that SOM is rather irrelevant for this study since it is hydrophilic and hence would activate regardless under supersaturated conditions, but the authors have to do a better job communicating this early on in the article and providing proof. Since only a few corner cases can be thought of where a hydrophobic, viscous coating similar to PE might be generated in the atmosphere, it seems correct to at least describe the expected behavior of the vast majority of organic material that is present in the atmosphere. Here, it would be worthwhile to briefly summarize the effects of these glassy, but hydrophilic, coatings to water uptake under subsaturated conditions, a research topic that has received much attention in the past (Mikhailov et al., 2009, Zobrist et al., 2011, Berkemeier et al. 2014, Pajunoja et al., 2015, to name a few studies that looked at water uptake [kinetics]). The authors discuss this issue very openly in Sect. 4, however, this discussion is absent from the introduction. The abstract reads especially strong: "it is concluded that mass transfer limitations by glassy organic shells are unlikely to affect cloud droplet activation near laboratory temperatures". As long as the authors do not show experiments with atmospherically more relevant material, this conclusion cannot stand by itself (also, p.9, l.17 needs a reference).

The authors describe their experimental and theoretical approach in detail. However, could the authors, in simple words, recite from their 2016 paper (Rothfuss and Petters, 2016) how they infer the slope of the viscosity vs temperature curve from a coalescence experiment, which to me seems to be more like a point measurement of viscosity?

The authors make a reasonable case that the PE/AS dimer particles should exhibit core-shell morphology (e.g. change in mobility diameter), but also point out that they have no exact proof for that. Since these doubts are communicated clearly, I can concur with their approach.

The authors give an explanation why the activated fraction for the core-shell particles never reaches unity (Figs. 4 and 5), a curious observation since without the thermal conditioning step, the activated fraction seems to go a little higher (at least at 0.62 % water supersaturation, Fig. 4). While their explanation seems logical overall, can they exclude the possibility that some of these 10-20 % of particles are PE/AS particles and don't activate because they are the only ones having a "true" core-shell morphology (not cracked, not partially-engulfed)? In general, the fitted line shown here is somewhat misleading since it doesn't account for the fact that activation does not reach unity.

Hence, despite few, small issues, I find this work very publishable and an interesting data point for the relevance of organic coatings for cloud activation. I can recommend it for publication in ACP after minor revision.

**References**

Berkemeier, T., Shiraiwa, M., Pöschl, U., and Koop, T.: Competition between water uptake and ice nucleation by glassy organic aerosol particles. Atmos. Chem. Phys., 14(22), 12513–12531, (2014).

Mikhailov, E., Vlasenko, S., Martin, S. T., Koop, T., and Pöschl, U.: Amorphous and crystalline aerosol particles interacting with water vapor: conceptual framework and experimental evidence for restructuring, phase transitions and kinetic limitations. At-

mos. Chem. Phys., 9(24), 9491–9522, (2009).

Pajunoja, A., Lambe, A. T., Hakala, J., Rastak, N., Cummings, M. J., Brogan, J. F., . . ., Virtanen, A.: Adsorptive uptake of water by semisolid secondary organic aerosols. Geophys. Res. Lett., 42(8), 3063–3068, (2015).

Rothfuss, N. E., and Petters, M. D.: Coalescence-based assessment of aerosol phase state using dimers prepared through a dual-differential mobility analyzer technique. Aerosol Sci. Technol., 50(12), (2016).

Zobrist, B., Soonsin, V., Luo, B. P., Krieger, U. K., Marcolli, C., Peter, T., and Koop, T.: Ultra-slow water diffusion in aqueous sucrose glasses. Phys. Chem. Chem. Phys., 13(8), 3514, (2015).
* * *

---

## Referee Comment (RC2) · Anonymous Referee #1 · 3 Oct 2018

The study by Tandon et al. investigates whether an organic hydrophobic glassy coating influences the cloud condensation nuclei activity of ammonium sulfate particles. To do this, polyethylene particles and ammonium sulfate particles of opposite charge were separately produced and then coagulated. The morphology of the resulting charge neutral mixed particles was varied by heating. These experiments showed that coatings up to volume fractions of 97 % did not change the critical supersaturation of 50 nm ammonium sulfate particles. Assuming that heating changes the mixed particle morphology from partially to completely engulfed, the authors conclude that mass transfer

limitations by glassy organic shells are unlikely to affect cloud droplet activation near laboratory temperatures. The authors address here a relevant and timely topic of atmospheric research that is well suited for publication in ACP. The experiments are intelligently designed and well performed. However, I have a major concern regarding the interpretation of the results that needs to be addressed before publication. Namely, the authors are too confident that the morphology is core-shell. Whether core-shell or partially engulfed morphologies are adopted depends on the surface tensions of the two involved phases and the interfacial tension between them. Estimating the spreading coefficient allows to predict which of these two morphologies is thermodynamically favored (Krieger et al., 2012; Reid et al., 2011). Partially engulfed configurations can also appear almost spherical and be misinterpreted as core-shell. Therefore, the spreading coefficient for the polyethylene-ammonium sulfate system should be calculated to assess the most likely morphology. Core-shell morphologies prevail in liquid-liquid-phase separated systems consisting of an aqueous organic and an aqueous inorganic phase (e.g. Ciobanu et al., 2009; Song et al., 2012; Stewart et al., 2015), while partially engulfed morphologies were observed for mixtures of hydrocarbons with NaCl, systems that come close to the one investigated in the present study. The authors showed that a hydrophobic coating does not impede CCN activation, however, most probably not because of lack of mass transfer limitations by glassy shells but due to an incomplete coating given by a partially engulfed morphology. This point needs to be adequately addressed before publication. The authors should reconsider their interpretation. They need to explain why their system should not relax into the thermodynamic favored state of partially engulfed.

Specific comments:

The expression "dimers" is used to refer to mixed polyethylene-ammonium sulfate particles. The definition of "dimer" in the online Oxford dictionary is: "A molecule or molecular complex consisting of two identical molecules linked together." The Collins English Dictionary defines: "a molecule composed of two identical simpler molecules

(monomers)." In the present study, "dimer" is used for particles obtained by coalescing two chemically very diverse particles. The authors should refrain from using "monomers" and "dimers" and search for alternative expressions.

Page 1, line 14: "polyethylene is taken as proxy for hydrophobic glass organic material". PE has a C:O ratio of zero. This is too low for a good proxy of atmospheric hydrophobic material.

Page 4, line 7 - 13: how was the morphology determined? Partially engulfed particles might appear almost spherical and still, the engulfed phase is in contact with the gas phase.

Page 7, lines 24 – 32: Here it is explicitly stated that the exact morphology could not be determined experimentally. Therefore, the authors should infer it from thermodynamic considerations using the spreading coefficient.

Page 8, lines 24: "core/shell" should be replaced by "spherical".

Page 10, lines 1 – 13: long chain fatty acids are a completely different case than PE, because they are surface active and should therefore fully cover the AS surface. Therefore, fatty acid coatings can very well hinder water transfer while a PE coating does not.

Page 10, lines 17 – 19: As stated above PE is not a good surrogate for high molecular weight weakly functionalized hydrocarbons: atmospherically relevant hydrophobic organics usually carry more double bonds and more aromatic rings. Moreover, the O:C ratio is zero for PE, which is hardly found in ambient organic aerosols.

Page 10, lines 29 – 31: It is not necessary to invoke fissures because the surface tension forces will NOT lead to a completely engulfed core with uniform coating. See my general comment.

Page 10, line 19 – 20: Do you have any indication that the carbon number may have changed?

Page 11. Lines 10 – 29: These conclusions need to be rewritten: With the experiments performed with PE, water transfer limitations and a low mass accommodation coefficient cannot be ruled out in the case of fatty acid coatings. If a hydrophobic organic mixture contains a share of surface active species, it might very well hinder CCN activation. The type of experiment performed in this study needs to be repeated with fatty acid containing hydrocarbons to come to a conclusion.

Page 12, lines 23 – 25: "Potential explanations are cracks formed during annealing, non-uniform coating thickness, or fast diffusion of small molecules through polymer membranes. It is argued that processes that may form glassy hydrophobic organic shells on atmospheric particles will result in similar imperfect shielding of hygroscopic cores." The explanation is a partially engulfed morphology. As stated above, this explanation might not hold for a fatty acid coatings. Therefore, this conclusion is not valid and needs to be removed.

Page 21, figure caption of Fig. 2: what is the definition of the "mean shape factor" ? It has not been defined in the main text. Is it the same as the "geometry factor"?

Page 21, figure caption of Fig. 2: what is the definition of "viscosity temperature" ? Again, this has not been defined in the main text?

Technical corrections: Page 2, line 4: maybe better: "small gas molecules" instead of small gases Page 5, line 14: "where" instead of "were" Page 8, line 7: "charged" instead of "charge".

References: Ciobanu, V. G., Marcolli, C., Krieger, U. K., Weers, U., and Peter, T.: Liquid-liquid phase separation in Mixed Organic/Inorganic Aerosol Particles, J. Phys. Chem. A, 113, 10966–10978, doi:10.1021/Jp905054d, 2009.

Krieger, U. K.; Marcolli, C.; Reid, J. P. Exploring the Complexity of Aerosol Particle Properties and Processes using Single Particle Techniques. Chem. Soc. Rev. 2012, 41, 6631−6662.

[Figure]

Reid, J. P.; Dennis-Smither, B. J.; Kwamena, N.-O.; Miles, R. E. H.; Hanford, K. L.; Homer, C. J. The Morphology of Aerosol Particles Consisting of Hydrophobic and Hydrophilic Phases: Hydrocarbons, Alcohols and Fatty Acids as the Hydrophobic Component. Phys. Chem. Chem. Phys. 2011, 13, 15559−15572.

Song, M., Marcolli, C., Krieger, U. K., Zuend, A., and Peter, T.: Liquid-liquid phase separation and morphology of internally mixed dicarboxylic acids/ammonium sulfate/water particles, Atmos. Chem. Phys., 12, 2691–2712, doi:10.5194/acp-12-2691-2012, 2012.

Stewart, D. J., Cai, C., Nayler, J., Preston, T. C., Reid, J. P., Krieger, U. K., Marcolli, C., and Zhang, Y. H.: Liquid-liquid phase separation in mixed organic/inorganic single aqueous aerosol droplets, J. Phys. Chem. A, 119, 4177–4190, 2015.

---

## Author Comment (AC1) · 20 Jan 2019

**Response to Reviewer Comments**

*January 19, 2019*

**Author Statement**

We thank both referees for their time to review this manuscript as well as their constructive critiques and suggestions. Both referees write that the manuscript is suitable for publication in ACP. The two major suggestions were (1) to expand the introduction to include material that was only presented in the discussion (referee #1), and (2) to explicitly discuss the thermodynamically favored morphology (referee #2). The requested material has been added to the manuscript. Below are itemized responses explaining how all of the referees' critiques were addressed.

**Response to Reviewer #1**

**Overview (Referee)**

In the study at hand, Tandon and co-workers investigate the role of hydrophobic organic material on the cloud condensation ability of ammonium sulfate (AS) particles by coating them with polyethylene (PE). In this careful study with innovative design, no effect of the coatings was observed, rendering hydrophobic organic coatings as uninfluential under the investigated conditions. The paper is well-written, the topic is of high relevance to the Atmospheric Chemistry and Physics community and fits well within the scope of Atmospheric Chemistry and Physics. [...]. Hence, despite few, small issues, I find this work very publishable and an interesting data point for the relevance of organic coatings for cloud activation. I can recommend it for publication in ACP after minor revision.

**Comments**

[1] Regarding the title, I have two issues with the use of "internally mixed" here.                    [1] Referee

1. This paper really rather looking at the effect of organic coatings on inorganic particles (admittedly, by creating an internally mixed particle) instead of the effect of organic coating on internally mixed particles.

2. While it is a set phrase in the aerosol community, it might paint the picture of hydrophobic substances being "mixed in" with inorganic substances.

There are not big issues, but the title threw me off slightly in the beginning and I wonder if this addition is needed.

[2] *We changed the title. In addition to the point raised by the referee, "internally mixed"*    [2] *Response*
*should really be applied to a population of particles, while this paper discusses the potential*
*role of morphology in single particles.*

[3] **(TITLE) The effect of hydrophobic glassy organic material on the cloud conden-**    [3] **Revision**

**sation nuclei activity of  particles with different  morphologies**

[4]The biggest critique for the relevance of this work must be, as the author's state themselves in their discussion, the relevance of PE coatings for atmospheric particles. PE is indeed very different to the most common organic material in the atmosphere, Secondary Organic Material (SOM). The authors argue that SOM is rather irrelevant for this study since it is hydrophilic and hence would activate regardless under supersaturated conditions, but the authors have to do a better job communicating this early on in the article and providing proof. Since only a few corner cases can be thought of where a hydrophobic, viscous coating similar to PE might be generated in the atmosphere, it seems correct to at least describe the expected behavior of the vast majority of organic material that is present in the atmosphere. Here, it would be worthwhile to briefly summarize the effects of these glassy, but hydrophilic, coatings to water uptake under subsaturated conditions, a research topic that has received much attention in the past (Mikhailov et al., 2009, Zobrist et al., 2011, Berkemeier et al. 2014, Pajunoja et al., 2015, to name a few studies that looked at water uptake [kinetics]). The authors discuss this issue very openly in Sect. 4, however, this discussion is absent from the introduction.

[4] Referee

[5]*The introduction was revised as follows.*

[5] *Response*

[6]**(INTRODUCTION) Atmospheric aerosol sometimes consists of mixed particles with inorganic and organic components present in comparable mass fractions (Cruz and Pandis, 2000; Pöschl et al., 2010; Prenni et al., 2003). Most of the inorganic compounds, commonly salts, are hygroscopic. The organic fraction is composed of a large number of compounds, many of which remain unspeciated. The contribution of the organic fraction to cloud condensation nuclei (CCN) activity depends on their hygroscopicity. The effective hygroscopicity parameter (Petters and Kreidenweis, 2007) of organic compounds varies between 0 and ~0.3, which is based on laboratory, field, and modeling studies (e.g. Prenni et al. 2007; Gunthe et al. 2009; Mikhailov et al., 2009; Chang et al. 2010; Massoli et al. 2010; Cappa et al. 2011; Mei et al. 2013; Pajunoja et al., 2015; Petters et al. 2016; Nakao 2017).  A  small subset of species constituting the organic fraction, e.g. long chain n-alkanes with carbon number 16 and above and their fatty acids, have very low water affinity (Jacobson et al., 2000; Jimenez et al., 2009; Petters et al., 2016). The organic fraction can thus be divided into hydrophobic and hydrophilic compounds. Hydrophobic compounds do not take up water and do not contribute to CCN activity. Particles containing both organic and inorganic compounds may assume different phase states and morphologies.**

**The phase state of dry organic matter may be crystalline, amorphous solid or semisolid, or liquid. When formed from drying of aqueous solutions droplets,**

[6] **Revision**

the organic phase often partially or fully encapsulates the inorganic core (e.g. Ciobanu et al., 2009; Freedman et al., 2010; Reid et al., 2011; Krieger et al., 2012; Stewart et al., 2015; Altaf and Freedman, 2017). Bigg (1986) first suggested that organic compounds located at the particle surface may kinetically retard water uptake and thus reduce the particle's ability to promote CCN activity. This reduction in CCN activity may occur through several pathways. First, the mass accommodation coefficient of water may be reduced (e.g. Chuang et al., 1997; Nenes et al., 2001). Second, slow dissolution kinetics may reduce the number of dissolved molecules in the aqueous solution (Asa-Awuku and Nenes, 2007; O'Meara et al., 2016). Finally, high viscosity may slow diffusion of water (e.g. Zobrist et al., 2011) and thus may affect hygroscopic growth under subsaturated conditions (Mikhailov et al., 2009), especially at cold temperature (Berkemeier et al., 2014; Petters et al., 2019). Measurements of the diffusion coefficient of water through hydrophilic viscous matrices (Price et al., 2015), measurements of the equilibration time scale of glassy hydrophilic organic compounds with water at high RH (Rothfuss et al., 2018), and measurements of the interplay between viscosity, hygroscopic growth, and CCN activity (Pajunoja et al., 2015) suggest that hydrophilic liquid or hydrophilic amorphous solid organic matrices are unlikely to hinder droplet activation. The main reason for this behavior is that small amounts of water uptake plasticize the organic phase, which in turn removes diffusion limitations. Plasticization, however, does not occur in hydrophobic organic substances, which therefore are more plausible candidate compounds to kinetically limit water uptake and droplet activation.

Laboratory studies have tested the influence of hydrophobic organic coatings on hygroscopic growth and CCN activity (Abbatt et al., 2005; Cruz and Pandis, 1998; Garland et al., 2005; Hämeri et al., 1992; Nguyen et al., 2017; Ruehl and Wilson, 2014; Xiong et al., 1998). ...

The abstract reads especially strong: "it is concluded that mass transfer limitations by glassy organic shells are unlikely to affect cloud droplet activation near laboratory temperatures". As long as the authors do not show experiments with atmospherically more relevant material, this conclusion cannot stand by itself.

[7]*The conclusion is based on the synthesis of literature data (which was included in the original statement) and the data presented here. We weakened the statement from "concluded" to "suggested".*    [7] *Response*

[8]**Based on these results, and a synthesis of literature data, it is  suggested that mass transfer limitations by glassy organic shells are unlikely to affect cloud droplet activation near laboratory temperatures.**    [8] *Revision*

[9](also, p.9, l.17 needs a reference)    [9] Referee

[10] *Response*

**[11] The two principle mechanisms of the delay are poor mass accommodation and diffusional limitation through the organic matrix (e.g. Chuang et al., 1997, Nenes et al., 2001, Zobrist et al., 2011). Either effect also implies some influence of particle morphology on CCN activation**

[11] **Revision**

[12] The authors describe their experimental and theoretical approach in detail. However, could the authors, in simple words, recite from their 2016 paper (Rothfuss and Petters, 2016) how they infer the slope of the viscosity vs temperature curve from a coalescence experiment, which to me seems to be more like a point measurement of viscosity?

[12] Referee

[13] *The observed viscosity range is shown by the gray circles in the right panels of Figure 2. The highlighted text was inserted to the manuscript to clarify the range in the methods section.*

[13] *Response*

**[14] The shape factor is converted to viscosity, $\eta$ via a lookup table of modified Frenkel sintering theory (Pokluda et al., 1997; Rothfuss and Petters, 2016). Conditioner residence time, monomer diameter, and surface tension assumed in the conversion are $t$ = 5 s, $D_{mono}$ = 50 nm, and $\sigma$ = 0.034 J m$^{-2}$ , respectively. The assumed surface tension is based on measured values for polyethylene melts (Dettre and Johnson, 1966; Owens and Wendt, 1969). The inferred viscosity is only weakly dependent on the assumed surface tension value (Marsh et al., 2018; Rothfuss and Petters, 2016). Particle shape factors vary between $1 \leq \xi \leq 4$, which map to a narrow range of viscosity between $10^{-5}$ Pa s and $3 \times 10^{-6}$ Pa s for the selected conditioner residence time, monomer diameter, and surface tension. The inferred temperature dependent viscosity over this range is fitted to a modified Vogel-FulcherTammann (VFT) equation (Rothfuss and Petters, 2017).**

[14] **Revision**

[15] The authors make a reasonable case that the PE/AS dimer particles should exhibit core-shell morphology (e.g. change in mobility diameter), but also point out that they have no exact proof for that. Since these doubts are communicated clearly, I can concur with their approach. The authors give an explanation why the activated fraction for the core-shell particles never reaches unity (Figs. 4 and 5), a curious observation since without the thermal conditioning step, the activated fraction seems to go a little higher (at least at 0.62 % water supersaturation, Fig. 4). While their explanation seems logical overall, can they exclude the possibility that some of these 10-20 % of particles are PE/AS particles and don't activate because they are the only ones having a "true" core-shell morphology (not cracked, not partially-engulfed)? In general, the fitted line shown here is somewhat misleading since it doesn't account for the fact that activation does not reach unity.

[15] Referee

[16]*As stated in the text, our conjecture is that a small fraction of decharged monomers transmits through the electrostatic filter, wherefore the distribution contains some fraction of pure PE particles that will not activate. This conjecture is based on extensive characterization of particle transmission through the system, which are the topic of a separate publication that is in preparation for submission to Aerosol Science and Technology (Rothfuss et al., in prep.). However, we did not quantify monomore decharging during the experiment reported here. To prove that some particles are fully coated and others are not, we would need to have a single particle composition measurement to show that the aerosol passed to the CCN instrument is indeed externally mixed. We therefore cannot categorically exclude the possibility that a small fraction has a "true" core-shell morphology that inhibited activation. In response we revised the paragraph to allow for this possibility.*

[17]**In some experiments, the activated fraction did not approach unity at high supersaturation. For experiments with disparate diameter of AS and PE, e.g. 50 nm AS and 120 nm PE, the dimer diameter will approach that of the larger monomer. Since a small fraction of decharged monomers also transmits (Rothfuss and Petters, 2016, Petters, 2018), the distribution may contains some fraction of pure PE particles that will not activate. Thus, the lack of 100% activation at high SS is likely an artifact and not due to particle morphology. However, it is also possible that those particles are composed of PE/AS and have assumed a core-shell morphology that is sufficient to hinder activation.**

[16] *Response*

[17] **Revision**

*Response to Reviewer #2*

**Summary comments (Referee)**

The study by Tandon et al. investigates whether an organic hydrophobic glassy coating influences the cloud condensation nuclei activity of ammonium sulfate particles. To do this, polyethylene particles and ammonium sulfate particles of opposite charge were separately produced and then coagulated. The morphology of the resulting charge neutral mixed particles was varied by heating. These experiments showed that coatings up to volume fractions of 97% did not change the critical supersaturation of 50 nm ammonium sulfate particles. Assuming that heating changes the mixed particle morphology from partially to completely engulfed, the authors conclude that mass transfer limitations by glassy organic shells are unlikely to affect cloud droplet activation near laboratory temperatures. The authors address here a relevant and timely topic of atmospheric research that is well suited for publication in ACP. The experiments are intelligently designed and well performed.

**Major Comment**

[18]However, I have a major concern regarding the interpretation of the results that needs to be addressed before publication. Namely, the authors are too confident that the morphology is core-shell. Whether core-shell or partially engulfed morphologies are adopted depends on the surface tensions of the two involved phases and the interfacial tension between them. Estimating the spreading coefficient allows to predict which of these two morphologies is thermodynamically favored (Krieger et al., 2012; Reid et al., 2011). Partially engulfed configurations can also appear almost spherical and be misinterpreted as core-shell. Therefore, the spreading coefficient for the polyethylene-ammonium sulfate system should be calculated to assess the most likely morphology. Core-shell morphologies prevail in liquid-liquid-phase separated systems consisting of an aqueous organic and an aqueous inorganic phase (e.g. Ciobanu et al., 2009; Song et al., 2012; Stewart et al., 2015), while partially engulfed morphologies were observed for mixtures of hydrocarbons with NaCl, systems that come close to the one investigated in the present study. The authors showed that a hydrophobic coating does not impede CCN activation, however, most probably not because of lack of mass transfer limitations by glassy shells but due to an incomplete coating given by a partially engulfed morphology. This point needs to be adequately addressed before publication. The authors should reconsider their interpretation. They need to explain why their system should not relax into the thermodynamic favored state of partially engulfed.

[18] Referee

[19]*We thank the referee to push us to clarify what is and is not known about the formed particle morphology. Furthermore, it is true that partially engulfed particles may appear almost spherical and our technique is not yet sensitive enough to detect small differences in particle shape. This will be acknowledged in the revised text.*

*The theory and experimental observations cited by the referee are not directly relevant to the system here. The spreading coefficient model discussed in Reid et al. (2011) and Krieger et al. (2012) goes back to Torza and Mason (1970) and describes two immiscible liquid*

[19] *Response*

*drops suspended in a third immiscible liquid. Similarly, the cited observations refer to the equilibrium morphology of liquid-liquid phase separated systems on supermicron droplets.*

*The case here is different. At high temperature a liquid polyethlyene droplet is in contact with a solid 50 nm diameter ammonium sulfate core. The assumed morphology of a sessile droplet on a spherical solid substrate depends on the solid/liquid, solid/vapor, liquid/vapor tensions, as well as the tension of the three phase contact line (Iwamatsu, 2016). The line tension term gains importance with decreasing particle diameter. Iwamatsu (2016) provides theory to predict the equilibrium morphology of such a system, which can be (1) complete wetting (core-shell morphology), (2) cap shaped droplet (core-shell spherical morphology with uneven coating thickness), (3) cap shaped droplet (partially engulfed morphology), (4) complete drying (dumbbell shaped morphology).*

*We do not know all relevant tensions to predict the shape. We know the liquid/vapor tension of polyethlene melts (0.034 J m-2, Dettre and Johnson, 1966; Owens and Wendt, 1969) and we expect that the solid/vapor tension of ammonium sulfate exceeds 0.080 J m-2, the value of concentrated solutions with water (Petters and Petters, 2016). The latter assumption is based on the notion that the saturated solution can approximate the surface tension of the solid/vapor interface and the analogous solid/vapor interface for NaCl, where the solid/vapor tension slightly exceeds that of the liquid/vapor tension of the saturated solution (Bahadur et al., 2007). We do not know the liquid/solid tension, the line tension or Young's contact angle of liquid PE on ammonium sulfate. Thus a theoretical prediction is not possible. However, we offer some discussion that argues in favor of complete wetting, i.e. core-shell morphology.*

1.  *Assuming that line tension is neglible, Young's law applies*

$$\gamma_{sg} - \gamma_{sl} = \gamma_{lg} \cos \theta_Y$$

*where $\gamma_{sg}, \gamma_{sl}, \gamma_{lg}$ are the solid/gas, solid/liquid, and liquid/gas interfacial tensions and $\theta_Y$ is Young's contact angle. Wetting occurs as $\theta_Y$ approaches 0°. Given that $\gamma_{sg}$ of inorganic salts is large (here estimated > 0.08 J m-2) and $\gamma_{lg}$ is small for liquid PE (< 0.034 J m-2), wetting will occur even if $\gamma_{sl}$ were appreciable (< 0.046 J m-2). (A large $\gamma_{sl}$ implies repulsive forces between the substrate and the liquid). In general, a low liquid/vapor surface tension will favor wetting and the liquid/gas tension term often dominates.*

2.  *Wetting at nanonscale is more efficient than at microscale. The stability diagram given by Iwamatsu (2016, their Fig. 7) suggests that for a sessile droplet on a spherical solid substrate wetting even occurs for $\theta_Y > 0°$. The upper contact angle where complete wetting is observed varies. This implies that even stronger respulsive forces than implied in $\gamma_{sl}$ = 0.046 J m-2 could be overcome in nanoscale systems due to the line tension effect.*

3.  *In the absence of theoretical predictions, findings from the literature are presented to corroborate the conjecture that complete wetting by liquid PE is plausible.*

    (a) *Freedman et al. (2010) show that ammonium sulfate particles coated with palmitic acid (produced through a coating oven), remain in core-shell structure. Palmitic acid and liquid PE both have low liquid/vapor interfacial tension. Palmitic acid and PE*

*are both dominated by CH2 groups and thus $\gamma_{sl}$ for these two substances should be similar.*

(b) *Reid et al. (2011) argue that the assumption that core–shell morphologies are formed "is well founded for a liquid phase surrounding a solid crystalline core". The effect cited by the referee that partially engulfed morphologies are observed for mixtures of hydrocarbons with aqueous NaCl is, to our understanding, due to the lowering of the aqueous solution surface tension by the organic, which does not occur in the crystalline case.*

*Despite the above, we freely acknowledge that (1) near spherical particle shape does not preclude partially engulfed morphology, (2) that we do not have empirical proof of particle morphology, (3) that the possibility of partially engulfed morphology cannot be easily dismissed, and (4) that forces during annealing may alter the equilibrium morphology of the liquid PE/solid AS particle. We have added a paragraph that discusses the possible morphologies, and include partially engulfed equilibrium morphology in the list of possible explanations why not kinetic limitations were observed.*

[20]**Since the melting point of AS (280 °C, Lide and Haynes, 2009) is much higher than that of PE (107 °C) the liquid PE is expected to partially or fully engulf the AS core. The assumed morphology of a liquid droplet on a spherical solid substrate depends on the solid/liquid, solid/vapor, liquid/vapor tensions, as well as the tension of the three phase contact line (Iwamatsu, 2016). The equilibrium morphology of this system can be (1) core-shell morphology (complete wetting), (2) core-shell morphology with uneven coating thickness (cap shaped droplet), (3) partially engulfed morphology (cap shaped droplet), or (4) dumbbell shaped morphology (complete drying). Parenthetical terms correspond to those used by Iwamatsu (2016). The line tension term becomes increasingly important at smaller scales. Line tension drives the system toward core-shell morphology. There is insufficient information about the AS/PE system at 95 °C to theoretically evaluate the equilibrium morphology. However, the liquid/vapor tension of polyethylene melts is 0.034 J m-2 (Dettre and Johnson, 1966; Owens and Wendt, 1969) and in principle a low liquid/vapor tension will favor the spreading of liquid on a high-energy surface. Reid et al. (2011) argue that core–shell morphologies are favored for a liquid phase surrounding a solid crystalline core. Freedman et al. (2010) show that ammonium sulfate particles coated with palmitic acid (produced through a coating oven), remain in core-shell structure. Palmitic acid and liquid PE both have low liquid/vapor interfacial tension. Palmitic acid and PE are both dominated by CH2 functional groups and thus the solid/liquid tension and line tension terms for these two substances should be similar.**

**The particles are cooled down to system temperature after they relax into spherical shape in the temperature-controlled loop. Based on the above we argue that the final morphology is a core-shell structure. However, we cannot dismiss the possibility that particles assumed a partially engulfed morphology. That morphology may be either due to equilibrium shape or arise during the anneal-**

[20] Revision

ing of the particle.  Thus, whether the coating of annealed PE is of uniform thickness is unknown. Scanning Electron Microscopy with Energy Dispersive X-ray spectroscopy of PE/AS dimer particles with larger diameters was attempted. However, the analysis was inconclusive due to insufficient resolution resulting from poor conductivity of the prepared samples and necessary limits in maximum electron beam intensity to ensure that the PE did not melt.

**Specific Comments**

[21]Page 1, line 14: polyethylene is taken as proxy for hydrophobic glass organic material. PE has a C:O ratio of zero. This is too low for a good proxy of atmospheric hydrophobic material.

[21] Referee

[22]*This has been addressed in response to comment* [37]*Referee (see below).*

[22] *Response*

[23]The expression "dimers" is used to refer to mixed polyethylene-ammonium sulfate particles. The definition of "dimer" in the online Oxford dictionary is: "A molecule or molecular complex consisting of two identical molecules linked together." The Collins English Dictionary defines: "a molecule composed of two identical simpler molecules (monomers)." In the present study, "dimer" is used for particles obtained by coalescing two chemically very diverse particles. The authors should refrain from using "monomers" and "dimers" and search for alternative expressions.

[23] Referee

[24]*The usage of monomer and dimer in the context of our method has not yet been critiqued by any of the referees of five prior publications. Google search results for "monomer particle" and "dimer particle" show 7760 and 2200 results, which appear predominately from the scientific literature, including the aerosol literature. We believe that meaning of the terms monomer and dimer in the manuscript is clear. Nonetheless, we have also added the clarifying particle to the abstract and include a strict definition in the text.*

[24] *Response*

[25](ABSTRACT) ... charge-neutral polyethylene-ammonium sulfate dimer particles were then isolated for online observation. Morphology of these dimer particles was varied by heating, such that liquefied polyethylene partially or completely engulfed the ammonium sulfate.

[25] Revision

(INTRODUCTION) ... is used to generate mixed particles composed of PE/PE and PE/AS dimer particles. Viscosity and CCN activity of pure PE particles is quantified. Dimer particle morphology

**(METHODS) The basic methodology for dimer particle preparation has been presented in detail elsewhere (Marsh et al., 2018; Rothfuss and Petters, 2016). Briefly, two monodisperse particle populations of opposite charge are generated using differential mobility analyzers (DMAs) operated with positive and negative polarity power supplies. The streams are merged and particles are given time to coagulate. Coagulated dimers particles from particles carrying +1/−1 charge are charge neutral and transmit through an electrostatic filter. The terms "monomer particle" or monomer are used to refer to particles transmitted by a single DMA and the term "dimer particle" or dimer are used to refer coagulated particles. These particles are available for further manipulation and measurement.**

[26]Page 4, line 7 - 13: how was the morphology determined? Partially engulfed particles might appear almost spherical and still, the engulfed phase is in contact with the gas phase.

[26] Referee

[27]*Morphology is only determined indirectly via change in drag force. It is therefore correct that partially engulfed might appear almost spherical and this is included in the revision. (Please see response to the major comment.)*

[27] *Response*

[28]Page 7, lines 24 – 32: Here it is explicitly stated that the exact morphology could not be determined experimentally. Therefore, the authors should infer it from thermodynamic considerations using the spreading coefficient.

[28] Referee

[29]*Please see response to major comment.*

[29] *Response*

[30]Page 8, lines 24: "core/shell" should be replaced by "spherical".

[30] Referee

[31]*Please see response to major comment.*

[31] *Response*

[32]Page 10, lines 1 – 13: long chain fatty acids are a completely different case than PE, because they are surface active and should therefore fully cover the AS surface. Therefore, fatty acid coatings can very well hinder water transfer while a PE coating does not.

[32] Referee

[33]*Please see our response to major comment.*

[33] *Response*

[34]Page 10, line 19 – 20: Do you have any indication that the carbon number may have changed?

[34] Referee

[35]*We do not but thermal decomposition of polymers is a possibility that deserves to be*

[35] *Response*

*mentioned. We revised as follows.*

[36]**The carbon number may have changed during heating via thermal decomposition . Whether or not this occurred cannot be determined from the available data. Regardless, the obtained PE particles had an estimated viscosity of $10^{12}$ Pa s at room temperature.**

[36] **Revision**

[37]Page 10, lines 17 – 19:  As stated above PE is not a good surrogate for high molecular weight weakly functionalized hydrocarbons:  atmospherically relevant hydrophobic organics usually carry more double bonds and more aromatic rings. Moreover, the O:C ratio is zero for PE, which is hardly found in ambient organic aerosols.

[37] Referee

[38]*We added this caveat.*

[38] *Response*

[39]**The exact carbon number of the PE particles here is not important, addition or subtraction of a few CH2 molecules does not significantly alter viscosity or hygroscopity (Petters et al., 2016; Rothfuss and Petters, 2017). Note, however, that atmospherically relevant hydrophobic organics may have double bonds, include aromatic rings (e.g. polycyclic aromatic hydrocarbons), and may include some functional groups other than $CH_x$. the PE particles used here are  used as a proxy for hydrophobic and glassy compounds.**

[39] **Revision**

[40]Page 10, lines 29 – 31:  It is not necessary to invoke fissures because the surface tension forces will NOT lead to a completely engulfed core with uniform coating. See my general comment.

[40] Referee

[41]*Please see response to general comment.*

[41] *Response*

[42]Page 11. Lines 10 – 29:  These conclusions need to be rewritten: With the experiments performed with PE, water transfer limitations and a low mass accommodation coefficient cannot be ruled out in the case of fatty acid coatings. If a hydrophobic organic mixture contains a share of surface active species, it might very well hinder CCN activation. The type of experiment performed in this study needs to be repeated with fatty acid containing hydrocarbons to come to a conclusion.

[42] Referee

[43]*CCN experiments with hydrophobic organic materials of low viscosity, including saturated and unsaturated fatty acids have been performed extensively in the literature (Cruz and Pandis 1998; Abbatt et al., 2005; Nguyen et al., 2017; Forestieri et al., 2018). These are discussed in the introduction and show with the exception of thickly coated palmitic acid that no hindrance in CCN activation was observed. Even the most favorable case, NaCl with > 60% palmitic acid (the referees hydrophobic surface active organic), changed the net*

[43] *Response*

*kappa only slightly. There are only two out of many data points in Nguyen et al. (2017, their Fig. 6) that show this behavior, and it is not entirely clear how measurement errors and uncertainties about the assumptions in the calculation (e.g. change in particle shape, evaluation of coating thickness) would factor into their closure calculation between predicted and observed kappa. Nguyen et al. (2017) write*

> *"For the saturated palmitic acid, a slight negative deviation (smaller observed kappa value than predicted) from the kappa addition rule was observed at thick coatings. It seems likely that this is due to limited diffusion of water through a solid coating as discussed above."*

*We don't disagree with that statement. However, the overwhelming number of data points in Nguyen et al. (2017) and all data from the remaining studies show no effect of fatty acid coatings on CCN activity. Given that the fatty acid/NaCl system has been extensively studied, we disagree with the referee that we need to repeat these experiments to factor it into a conclusion.*

*As to the broader point, the paragraph in question essentially argues that it is almost impossible to demonstrate kinetic limitations of in the lab. It is difficult to imagine processes that lead to more effective coatings in the atmosphere when compared to the laboratoru. We still believe that this conclusion is justified. However, the statement was revised to include partially engulfed morphologies and weakened the conclusion to the "majority of cases".*

[44]**Regardless, the imagined processes producing hydrophobic glassy coatings undergo drying and cooling cycles that will be susceptible to the same  issues reported here: cracks non-uniform coating thickness formed during drying or annealing, partially engulfed equilibrium morphology, and faster than expected diffusion through hydrocarbon films. We therefore conclude that mass transfer limitation by glassy organic shells is unlikely to affect cloud droplet activation in the majority of cases  Extension of this result to temperatures in the upper free troposphere where low temperature slows diffusion may require further experimentation.**

[44] Revision

[45]Page 12, lines 23 – 25: "Potential explanations are cracks formed during annealing, non-uniform coating thickness, or fast diffusion of small molecules through polymer membranes. It is argued that processes that may form glassy hydrophobic organic shells on atmospheric particles will result in similar imperfect shielding of hygroscopic cores." The explanation is a partially engulfed morphology. As stated above, this explanation might not hold for a fatty acid coatings. Therefore, this conclusion is not valid and needs to be removed.

[45] Referee

[46]*Please see our response to the major comment regarding partially engulfed morphology. It is stated in the introduction that the evidence is scant that fatty acid coatings hinder activation. However, we agree that it worthwhile adding a qualifying statement that thickly coated stearic acid may lead to hindrance.*

[46] Response

[47]**Potential explanations are cracks formed during annealing, non-uniform coat-**

[47] Revision

**ing thickness, formation of partially engulfed morphologies, or fast diffusion of small molecules through polymer membranes. It is argued that processes that may form glassy hydrophobic organic shells on atmospheric particles will result in similar imperfect shielding of hygroscopic cores. However, particles with thick coatings of some, but not all fatty acids are an exception to the preceding claim (Abbatt et al., 2005, Nguyen et al., 2017, Forestieri et al., 2018).**

[48] Page 21, figure caption of Fig. 2: what is the definition of the "mean shape factor"? It has not been defined in the main text. Is it the same as the "geometry factor"?

[48] Referee

[49] *Corrected.*

[49] *Response*

[50] **(METHODS) Next, the fitted mode diameters are binned into 3 K intervals (Marsh et al., 2018)**  **the mean geometry factor is derived for each bin. These data are then fitted to a logistic curve** [51] **Figure 2. Left plots: shape relaxation data for PE/AS (a) and PE/PE (c) 50nm/50nm dimers. Symbols are the mean geometry factor derived from the temperature binned mode diameter peaks.**

[50] **Revision**

[51] **Revision**

[52] Page 21, figure caption of Fig. 2: what is the definition of "viscosity temperature"? Again, this has not been defined in the main text?

[52] Referee

[53] *This was poor wording.*

[53] *Response*

[54] **Right plots:  temperature dependence of viscosity for PE/AS (b) and PE/PE (d) experiments.**

[54] **Revision**

[55] Technical corrections: Page 2, line 4: maybe better: "small gas molecules" instead of small gases Page 5, line 14: "where" instead of "were" Page 8, line 7: "charged" instead of "charge".

[55] Referee

[56] *Corrected.*

[56] *Response*

**References Cited**

Highlighted references were added to the manuscript. Non-highlighted references are used in response to referees. References that appeared in the original manuscript are cited there.

Bahadur, R. Russell, L. M., Alavi, S.: Surface Tensions in NaCl-Water-Air Systems from MD Simulations, J. Phys. Chem. B, 111, 11989–11996, 2007.

[revised manuscript text omitted]

---

## Author Response (AR2)

*Author Statement*

We thank the referees who evaluated the revised version of the manuscript for their time and feedback. The remaining comment is addressed below.

**Referee Comment**

The authors have responded satisfactory to most comments. Most importantly, they have included the possibility of partially engulfed morphologies in the discussion of the results. However, they are inaccurate in their reference to Reid et al. (2011). They added the following sentence to the revised manuscript:

> "Reid et al. (2011) argue that core–shell morphologies are favored for a liquid phase surrounding a solid crystalline core."

This citation is taken out of the context and needs to be revised. Within the context the sentence reads:

> "This is to be contrasted with the widely held assumption that core–shell morphologies are formed. While this assumption may be well founded for a liquid phase surrounding a solid crystalline core, this assumption cannot be made for aerosols containing discrete hydrophobic and hydrophilic liquid domains."

The main topic of Reid et al. (2011) is indeed the morphologies of liquid-liquid phase separated particles. They investigated decane/aqueous sodium chloride droplets and found partially engulfed morphologies at high RH, in accordance with predictions based on surface and interfacial tensions. The morphology changed to core-shell when RH and the volume fraction of decane decreased. Given the larger size of PE compared with decane and the rather stronger salting-out effect of AS compared with NaCl, partially engulfed is the likely morphology for the PE/AS particles, however, the authors are right to point out that the line tension needs to be taken into account for smaller particles.

*We were fully aware of the context of the statement. We reiterate that in our case the relevant system is liquid organic phase (liquified PE) that rests on a crystalline core (ammonium sulfate). During the partitioning there is no water. Therefore, the partially engulfed structures that form in the case of two aqueous liquid phases is not a good comparison. The statement of Reid et al. (2011) is offered as one piece of evidence for the likelihood of core-shell structures. The observation of a Freedman et al. (2010) who studied the morphology of palmitic acid/ammonium sulfate is offered as a second piece of evidence for the likelihood of core-shell structures. We again emphasize that we do not have proof for the final morphology and that this is clearly stated in the manuscript. To address the referees concern we revised the text as follows:*

**Reid et al. (2011) argue that core–shell morphologies are favored for a liquid phase surrounding a solid crystalline core, as is the case for liquid PE spreading over crystalline AS. However, this assumption is often invalid for particles containing hydrophobic and hydrophilic liquid domains (Reid et al., 2011). Freedman et al. (2010) ...**

[revised manuscript text omitted]